# Chemical Composition of Various *Nepeta cataria* Plant Organs’ Methanol Extracts Associated with *In Vivo* Hepatoprotective and Antigenotoxic Features as well as Molecular Modeling Investigations

**DOI:** 10.3390/plants11162114

**Published:** 2022-08-14

**Authors:** Milena D. Vukić, Nenad L. Vuković, Milan Mladenović, Nevena Tomašević, Sanja Matić, Snežana Stanić, Filippo Sapienza, Rino Ragno, Mijat Božović, Miroslava Kačániová

**Affiliations:** 1Department of Chemistry, Faculty of Science, University of Kragujevac, Radoja Domanovića 12, 34000 Kragujevac, Serbia; 2Kragujevac Center for Computational Biochemistry, Department of Chemistry, Faculty of Science, University of Kragujevac, Radoja Domanovića 12, 34000 Kragujevac, Serbia; 3Department of Science, Institute for Information Technologies Kragujevac, University of Kragujevac, Jovana Cvijića bb, 34000 Kragujevac, Serbia; 4Department of Biology and Ecology, Faculty of Science, University of Kragujevac, Radoja Domanovića 12, 34000 Kragujevac, Serbia; 5Rome Center for Molecular Design, Department of Drug Chemistry and Technology, Faculty of Pharmacy and Medicine, Sapienza University of Rome, Piazzale Aldo Moro 5, 00185 Rome, Italy; 6Faculty of Science and Mathematics, University of Montenegro, Džordža Vašingtona bb, 81000 Podgorica, Montenegro; 7Institute of Horticulture, Faculty of Horticulture and Landscape Engineering, Slovak University of Agriculture in Nitra, Tr. A. Hlinku 2, 94976 Nitra, Slovakia; 8Department of Bioenergy, Food Technology and Microbiology, Institute of Food Technology and Nutrition, University of Rzeszow, 4 Zelwerowicza St., 35601 Rzeszow, Poland

**Keywords:** *Nepeta cataria*, phenolic compounds, Wistar rats, hepatoprotective activity, antigenotoxic activity, structure-based pharmacological studies

## Abstract

This report summarizes the chemical composition analysis of *Nepeta cataria* L. flower, leaf, and stem methanol extracts (FME, LME, SME, respectively) as well as their hepatoprotective and antigenotoxic features *in vivo* and *in silico*. Herein, Wistar rat liver intoxication with CCl_4_ resulted in the generation of trichloromethyl and trichloromethylperoxy radicals, causing lipid peroxidation within the hepatocyte membranes (viz. hepatotoxicity), as well as the subsequent formation of aberrant rDNA adducts and consequent double-strand break (namely genotoxicity). Examined FME, LME, and SME administered orally to Wistar rats before the injection of CCl_4_ exerted the most notable pharmacological properties in the concentrations of 200, 100, and 50 mg/kg of body weight, respectively. Thus, the extracts’ hepatoprotective features were determined by monitoring the catalytic activities of enzymes and the concentrations of reactive oxidative species, modulating the liver redox status. Furthermore, the necrosis of hepatocytes was assessed by means of catalytic activities of liver toxicity markers. The extracts’ antigenotoxic features were quantified using the comet assay. Distinct pharmacological property features may be attributed to quercitrin (8406.31 μg/g), chlorogenic acid (1647.32 μg/g), and quinic acid (536.11 μg/g), found within the FME, rosmarinic acid (1056.14 μg/g), and chlorogenic acid (648.52 μg/g), occurring within the LME, and chlorogenic acid (1408.43 μg/g), the most abundant in SME. Hence, the plant’s secondary metabolites were individually administered similar to extracts, upon which their pharmacology *in vivo* was elucidated in silico by means of the structure-based studies within rat catalase, as a redox marker, and rat topoisomerase IIα, an enzyme catalyzing the rat DNA double-strand break. Conclusively, the examined *N. cataria* extracts in specified concentrations could be used in clinical therapy for the prevention of toxin-induced liver diseases.

## 1. Introduction

*Nepeta* represents a large genus belonging to the Lamiaceae family, subfamily Nepetoideae, and tribe Mentheae [1]. This genus comprises about 300 species, the majority of which are aromatic plants [2,3]. Several species of this genus have been reported to possess medicinal properties such as diuretic, sedative diaphoretic, antispasmodic, antiasthmatic, expectorant, febrifuge, antitumor, antibacterial, antifungal, and emmenagogue [4,5,6,7]. Previous phytochemical analyses of *Nepeta* revealed the presence of several bioactive phytochemicals, such as polyphenols (phenolics and flavonoids), terpenoids [3,4,7,8], iridoid and iridoid glycosides [9,10,11]. Polyphenols, i.e., naturally occurring products of plants’ secondary metabolism, have shown a broad range of biological activities such as analgesic, anti-arthritis, antimicrobial, antipyretic, anti-inflammatory, hepatoprotective, and anti-thrombotic [12,13,14]. Owing to their potential therapeutic effects on health as well as their use in the human diet, these chemical entities have drawn significant attention within the scientific community [15,16,17]. More than twenty phenolic derivatives and thirty-five flavonoids and their derivatives have been isolated from diverse *Nepeta* species until now [3].

*Nepeta catar**ia* L. (catnip or catmint) has long been used in traditional medicine in France, England, and regions of North America, in the form of teas and infusions, for treating nervousness, anxiety, insomnia, and symptoms related to gastrointestinal upset, including dyspepsia, cramping, and meteorism. Moreover, it is associated with a diuretic effect preventing water retention, as well as for handling arthritis, coughs, hives, fevers, and viruses [18]. Due to these properties, the plant is widely used for its antispasmodic, expectorant, diuretic, antiseptic, and antiasthmatic effects [3,7,19]. An interesting application of *N. cataria* is its usage in pet toys [19], given that catnip alters behavior and produces pleasurable sensations in both wild and domestic cats, as well as in other mammals [20]. A study also associated dried leaves with pleasurable experiences of catnip in humans [21].

With the fact that many of the ingested or inhaled environmental toxicants, drugs, and nutrients suffer from the first-pass effect, the liver’s function can be altered by acute or chronic exposure to hazardous chemical entities [22,23]. The biological activities and therapeutic applications of plant extracts and purified secondary metabolites have attracted attention for decades [22,23]; nevertheless, toxicological, hepatoprotective, and DNA protective studies need to be further investigated. There is a growing interest in the hepatoprotective and antigenotoxic roles of either plant-derived extracts or their isolated substances [24]. Accordingly, within this report, the hepatoprotective and antigenotoxic effects of *N. cataria* plant methanolic extracts were evaluated on Wistar rats exposed to carbon tetrachloride (CCl_4_). To the best of the authors’ knowledge, such an investigation has never been reported. The *N. cataria* methanolic extracts were fully characterized and found to be associated with eighteen secondary metabolites: quinic acid, phenolic acids (or their derivatives), and flavonoid aglycones (or their glycosides), which were quantified by LC-ESI-MS/MS, and their effects were investigated in vivo and interpreted by means of in silico studies.

## 2. Results and Discussion

### 2.1. Chemical Composition of Methanol Extracts of N. cataria

In agreement with previously published qualitative composition of *N. cataria* aerial parts [25], seventeen phenolic compounds and the quinic acid (an intermediate in plant phenolics biosynthesis), were selected as standards (Table 1) for the quantitative investigation of flower (FME), leaf (LME), and stem (SME) methanol extracts (Figure 1A–C, respectively) by means of LC-ESI-MS/MS analysis. The compounds concentrations, expressed as μg of compound per g of extract, were determined from the standards’ calibrated curves, prepared by means of a double dilution method (50.00 to 0.00153 μg/mL). The results of the qualitative and quantitative analyses showed that all the used standards were found in the different aerial part plant extract samples, and no other unknown component was detected (Table 1 and Figure 1). 

#### 2.1.1. Flowers Methanol Extract (FME) of *N. cataria*

Within the flavonoids class, flavonol quercetin, alongside its three glycosides, quercitrin, quercetin-3-*O*-glucoside, and quercetin-3-*O*-rutinoside, were observed in the amounts of 88.97, 8406.31, 72.97, 86.55 mg/g, respectively. Notably, quercetin-3-*O*-rutinoside nearly reached the amount of its aglycone. Flavonols isorhamnetin and kaempferol were found in amounts of 107.53 and 90.03 mg/g, respectively, while kaempferol-3-*O*-glucoside was present in almost four-fold higher concentrations (409.21 mg/g) related to its aglycone. Flavone luteolin as well as its 7-O-glucoside were the minor constituents of the FME (6.12 and 7.64 mg/g, respectively). Among phenolic acids, chlorogenic acid and rosmarinic acid were identified in high concentrations (1647.32 and 326.15 mg/g, respectively). Other phenolic acids were present in the concentration range of 15.93 to 138.21 mg/g (Table 1, Figure 1A).

#### 2.1.2. Leaves Methanol Extract (LME) of *N. cataria*

Within the LME, kaempferol-3-*O*-glucoside and quercetin-3-*O*-glucoside were observed as major flavonoids (374.13 and 354.33 mg/g, respectively). Among the quantified flavonol aglycones, isorhamnetin, kaempferol, and quercetin were present in notable concentrations (175.34, 82.87 and 78.33 mg/g, respectively), while flavones luteolin and luteolin-7-*O*-glucoside were found in low amounts (4.13 and 5.43 mg/g, respectively). Of the phenolic acids, rosmarinic and chlorogenic acids were identified as most abundant (1056.14 and 648.52 mg/g, respectively), while related to distinct phenolic acids, *p*-hydroxybenzoic and protocatechuic acids were observed in relatively lower amounts (363.32 and 261.42 mg/g, respectively, Table 1 and Figure 1B).

#### 2.1.3. Stems Methanol Extract (SME) of *N. cataria*

Inside of the SME, kaempferol-3-*O*-glucoside was the most abundant flavonol (168.54 mg/g). In addition, quercetin, isorhamnetin, quercetin-3-*O*-rutinoside, and quercetin-3-*O*-glucoside were found in significant amounts (94.54, 117.24, 114.01 and 87.97 mg/g, respectively), while other flavonoids were identified ranging from 1.02 to 67.25 mg/g. Chlorogenic acid was identified in high concentrations (1408.43 mg/g). Other phenolic acids were found in the range of 30.32 to 250.54 mg/g. The results of quantitative analysis of phenolics present in stems showed that except for the case of accumulation of chlorogenic acid (as the main constituent), concentrations of other investigated compounds are lower than in flowers and leaves (Table 1, Figure 1C).

#### 2.1.4. Implications of *N. cataria* Extracts Chemical Composition on the Plant’s Activity

Generally, it could be anticipated that the physiological properties of FME could be ascribed to the most abundant compounds: quercitrin, chlorogenic acid, and quinic acid (Table 1 and Figure 1A). Analogously, rosmarinic acid and chlorogenic acid could be the main actors in the LME (Table 1 and Figure 1B). Regarding SME, chlorogenic acid was the dominant compound and consequently could be assumed as the compound mainly responsible to be associated with its bioactivity (Table 1 and Figure 1C). 

From a literature survey, it has been found that polyphenol-rich extracts showed antigenotoxic activity [26,27]. However, to reach the DNA, the medicinal plants’ active ingredients must pass the liver barrier for which the initial physiological response of an organ occurs in terms of hepatoprotective potential [28]. Hence, before any antigenotoxic activity, the hepatoprotective potential of a given extract should be evaluated by means of the antioxidant abilities of contained compounds (phenolic acids and flavonoids) either while counteracting the liver damage [29,30], interfering with the drug-metabolizing and repairing enzymes, or interacting with signaling molecules important for cell survival [31]. *Ficus gnaphalocarpa* with hepatoprotective *in vitro* activity, showed a strong effect of quercitrin in reducing the CCl_4_-induced disruption of human hepatoma HEPG2 cell lines through its ability in preventing liver cell death and leakage of lactate dehydrogenase into a medium [32]. Chlorogenic acid was observed in a high concentration in the sea cucumber extract and significantly reduced thioacetamide-induced liver injury in rats [33], while rosmarinic acid, as the major compound of *Perrila frutescens*, decreased *tert*-butyl hydroperoxide (*t*-BHP)-induced oxidative liver damage [34].

Therefore, further experiments were directed toward creating the physiological scenario of treating the adult Wistar rats (*Rattus norvegicus*, hereinafter labeled as *r* to be used as a prefix for all reported data) with CCl_4_, a hazardous chemical entity used to induce serious implications on both membrane and genome (Figure 1) and to reveal the initial hepatotoxic/hepatoprotective features (Table 2, Table 3, Table 4 and Table 5) of extracts and the found chemical constituents (Table 1, Figure 1), as well as the associated genotoxic/antigenotoxic features (Table 6, Table 7 and Table 8). Within all the experiments, the control group comprised experimental animals treated *i.p*. with virgin olive oil (herein commercially available, 1 mL/kg bwt), due to the ability of such a mixture to decrease liver damage in rats caused by CCl_4_ [35].

Hence, for determining the above-defined extracts hepatotoxic (Table 2 and Table 4: groups III–V) and genotoxic features (Table 6: groups III-V), they were orally administered to adult Wistar rats, at 200 mg/kg bwt, in comparison with intraperitoneally (*i.p.*) applied CCl_4_ (Table 2 and Table 4: groups II) or olive oil (Table 2 and Table 4: groups I). The pure secondary metabolites (Table 1, Figure 1) and their features were also investigated in parallel (Table 3 and Table 5). Furthermore, for elucidating the hepatoprotective (Table 2 and Table 4: groups VI–XIV) and/or antigenotoxic (Table 7: groups VI-XIV) effect, different concentrations (50, 100, and 200 mg/kg bwt) of extracts and the pure metabolites were administered 5 days before the hazardous agent application (Table 2, Table 4 and Table 7: groups II and I; and Table 8). 

### 2.2. Hepatotoxic and Hepatoprotective Features of N. cataria Flower (FME), Leaf (LME), and Stem (SME) Methanol Extracts 

The liver intoxication with CCl_4_ through *r*CYP2E1 (a member of *r*Cytochrome P-450 family) leads to hepatotoxic metabolite production and among them can be mainly listed the trichloromethyl (*r*CCl_3_^●−^) and trichloromethylperoxy (*r*CCl_3_OO^●−^) radicals (Figure 1) [36], which can further induce the peroxidation of hepatocytes membrane polyunsaturated fatty acids (Table 2 and Table 4: group II, Table 3 and Table 5), likely contributing to hepatotoxicity [37]. Therefore, *N. cataria* extracts were initially investigated for their impact on lipid peroxidation upon CCL_4_ administration [36]. Hence, *N. cataria* extract behavior in the hepatocyte membrane oxidative stress scenario was indirectly assessed through the catalytic activity of superoxide dismutase (*r*SOD), the concentration of the thiobarbituric acid-reactive substance (*r*TBARS), the catalytic activity of catalase (*r*CAT), and the concentration of reduced glutathione (*r*GSH) (Figure 1, Table 2). Furthermore, the radical-induced damage of the hepatocytes’ membrane was estimated through the catalytic activities of aspartate transaminase (*r*AST), alanine transaminase (*r*ALT), alkaline phosphatase (*r*ALP), and γ-glutamyltransferase (*r*γ-GT), whereas bile damage was assessed through the catalytic activity of *r*ALP and *r*γ-GT (Table 4 and Table 5).

#### 2.2.1. Hepatotoxic and Hepatoprotective Features of *N. cataria* Flower Methanol Extract (FME) 

##### The Hepatocytes Redox Status

Within the liver, toxic radicals such as *r*CCl_3_^●^ are likely oxidized by *r*CYP2E1-activated molecular oxygen to produce *r*CCl_3_OO^●^ radicals, leading to the formation of superoxide radicals (*r*O_2_^●−^) as well (Figure 1) [38]. The *r*SOD catalyzes the dismutation of *r*O_2_^●−^ into oxygen and hydrogen peroxide (Figure 1), representing the *in viv**o* redox defense mechanism, and any decrease in *r*SOD catalytic activity is associated with running oxidative stress [39]. The administration of CCl_4_ (Table 2: group II) caused a 2.19-fold decrease in *r*SOD catalytic activity compared to the olive oil (Table 2: group I), confirming the enzyme as a sensitive marker for CCl_4_-induced liver injury. Furthermore, the administration of *N. cataria* FME at 200 mg/kg (Table 2: group III) led to *r*SOD activation (to the 75.28% of the *r*SOD catalytic activity of the control, Table 2: group I), a result that could not be attributed to the hepatotoxicity (since the FME did not cause severe lipid peroxidation and other liver oxidative damages; see further discussion), but could be interpreted as a normal liver physiological response. External confirmation of a later postulate was received upon distinct extraction, which has notably less-pronounced cellular response related to CCl_4_ (i.e., caused a 1.65-fold increase in *r*SOD catalytic activity compared to Table 2: group I). Furthermore, the high increase in *r*SOD catalytic activity, in terms of FME’s administration before CCl_4_ in matching concentration (Table 2: group VIII), pushed in favor that the remedy acted in an apparent hepatoprotective fashion (60.60% of *r*SOD catalytic activity related to Table 2: group I, and 1.33-fold above Table 5: group II), exerting negligible higher hepatoprotective potential related to lower concentrations (see Table 2: groups VI and VII). 

In the liver hepatocytes membrane, the *r*CCl_3_^●^ and *r*CCl_3_OO^●^ radicals induce the formation of malondialdehyde (*r*MDA) (Figure 1), a major product of lipid peroxidation [28]. Therefore, the additional level of *N. cataria* extract hepatoprotective ability was determined by complexing the *r*MDA with thiobarbituric acid, resulting in the formation of *r*TBARS (Figure 1), whose increased concentration indicates membrane damage [40]. Immersed in the membrane and lowering the concentration of *r*TBARS in physiological conditions, the *N. cataria* FME individually expressed notable hepato-protectivity (Table 2: group III, only 57.53% of *r*TBARS concentration measured related to Table 2: group I), beyond comparison to the hepatotoxicity of CCl_4_ (Table 2: group II, 4.80-fold higher TBARS concentration related to Table 5: group I). Such intensive hepatoprotection was evident even against CCl_4_ (Table 2: group VIII, 69.41% of *r*TBARS concentration related to Table 2: group I, and 6.91-fold lower *r*TBARS concentration than within Table 5: group II). 

When CCl_4_-based intoxication ends with hydrogen peroxide, *r*CAT degrades the product to water and molecular oxygen (Figure 1), and the decrease in the catalytic activity of *r*CAT leads to oxidative stress in tissue [41]. Therefore, readily expected, CCl_4_ caused a moderate drop in the catalytic activity of *r*CAT related to olive oil (Table 2: group II vs. group I, 81.04% of the catalytic activity). It was anticipated that *N. cataria* FME could counteract the CCl_4_-based intoxication (Table 2: group III, 98.20% of the catalytic activity of *r*CAT related to Table 2: group I), which was confirmed by administering it directly against CCl_4_ (Table 2: group VIII vs. group II, 1.08-fold higher CAT catalytic activity).

**Table 2 plants-11-02114-t002:** Total protein content (*r*TP), catalytic activities, and concentrations of liver antioxidant enzymes in rats exposed to different doses of *N. cataria* extracts and CCl_4_.

Groups	*r*TP (g/L)	*r*SOD (U/mg)	*r*TBARS (nmol/mg)	*r*CAT (U/mg)	*r*GSH (mg/g)
I	^1^ 28.39 ± 0.14	5.38 ± 0.03	2.19 ± 0.02	131.33 ± 0.15	16.04 ± 0.12
II	27.07 ± 0.12	2.45 ± 0.05 *	10.51 ± 0.14 *	106.43 ± 0.3 *	6.39 ± 0.16 *
III	30.01 ± 0.17	4.05 ± 0.11 *^†^	1.26 ± 0.08 ^†^	128.97 ± 0.08 ^†^	13.22 ± 0.20 *^†^
IV	26.13 ± 0.23	4.29 ± 0.02 *^†^	2.39 ± 0.09 ^†^	127.57 ± 0.10 ^†^	13.52 ± 0.12 *^†^
V	25.90 ± 0.34	4.31 ± 0.06 *^†^	1.33 ± 0.08 *^†^	124.31 ± 0.15 *^†^	14.79 ± 0.13 *^†^
VI	27.61 ± 0.24	3.26 ± 0.06 *^†^	2.24 ± 0.02 ^†^	113.07 ± 0.23 *^†^	10.63 ± 0.14 *^†^
VII	34.19 ± 0.17	3.26 ± 0.08 *^†^	1.88 ± 0.03 ^†^	114.34 ± 0.22 *^†^	11.44 ± 0.02 *^†^
VIII	27.48 ± 0.41	3.28 ± 0.09 *^†^	1.52 ± 0.06 *^†^	120.93 ± 0.11 *^†^	12.01 ± 0.18 *^†^
IX	28.17 ± 0.28	3.41 ± 0.04 *^†^	2.81 ± 0.09 *^†^	113.65 ± 0.28 *^†^	12.18 ± 0.08 *^†^
X	25.23 ± 0.16	3.65 ± 0.02 *^†^	1.82 ± 0.07 ^†^	126.40 ± 0.13 ^†^	13.26 ± 0.06 *^†^
XI	28.88 ± 0.13	3.09 ± 0.07 *^†^	2.93 ± 0.05 *^†^	119.32 ± 0.19 *^†^	12.18 ± 0.03 *^†^
XII	27.78 ± 0.23	3.31 ± 0.08 *^†^	4.31 ± 0.07 *^†^	123.80 ± 0.16 *^†^	12.79 ± 0.21 *^†^
XIII	29.56 ± 0.16	3.11 ± 0.09 *^†^	5.08 ± 0.10 *^†^	115.25 ± 0.20 *^†^	11.61 ± 0.13 *^†^
XIV	27.70 ± 0.32	3.25 ± 0.01 *^†^	4.99 ± 0.05 *^†^	124.34 ± 0.14 *^†^	12.18 ± 0.09 *^†^

I, Control group, animals treated orally for five days with distilled water and then intraperitoneally (*i.p*.) injected with 1 mL/kg body weight (bwt) in olive oil; II, CCl_4_ 1 mL/kg *i.p*.; III, *N.cataria* FME 200 mg/kg; IV, *N.cataria* LME 200 mg/kg; V, *N.cataria* SME 200 mg/kg; VI, *N. cataria* FME 50 mg/kg+CCl_4_; VII, *N. cataria* FME 100 mg/kg+CCl_4_; VIII, *N. cataria* FME 200 mg/kg+CCl_4_; IX, *N. cataria* LME 50 mg/kg+CCl_4_; X, *N. cataria* LME 100 mg/kg+CCl_4_; XI, *N. cataria* LME 200 mg/kg+CCl_4_; XII, *N. cataria* SME 50 mg/kg+CCl_4_; XIII, *N. cataria* SME 100 mg/kg+CCl_4_; XIV, *N. cataria* SME 200 mg/kg+CCl_4_; ^1^ Values represent mean ± SEM from three independent experiments; n = 5 rats per group; * *p* < 0.05 when compared with the negative control group; ^†^
*p* < 0.05 when compared with the CCl_4_ control group. Results are presented as equivalents of total protein concentration.

Further evidence that *N. cataria* FME was endowed with hepatoprotective activity was assessed by monitoring *r*GSH, a dual-mode antioxidant against reactive oxygen species emerging from CCl_4_ [38]. The production of *r*CCl_3_OO^●^ radicals and their interaction with membrane lipids may cause lipid radical formation and the production of *r*CCl_3_OOH, which is reduced to *r*CCl_3_OH by *r*GSH (Figure 1) [38,42]. Furthermore, hydrogen peroxide excess may be neutralized by *r*GSH (Figure 1) supporting the antioxidative features of *r*CAT [38]. Compared to CCl_4_ (Table 2: group II, 2.51-fold lower *r*GSH concentration related to Table 2: group I), the *N. cataria* FME did not exert hepatotoxicity (Table 2: group IV, 82.42% of the *r*GSH concentration measured in Table 2: group I). Similarly, while monitoring the *r*TBARS, the administration of *N. cataria* FME in the concentration of 200 mg/kg bwt before CCl_4_ (Table 2: groups VI, VII, and VIII) correlated with hepatoprotective effects (74.88% of *r*GSH concentration than within Table 2: group I and 1.88-fold higher *r*GSH concentration than within Table 2: group II were detected). The most abundant secondary metabolites of *N. cataria* FME (Table 1, Figure 1), quercitrin, chlorogenic acid, and quinic acid, showed to have an important role in the hepatoprotective activity against administered CCl_4_ (Table 3), supporting the previous discovery that hepatotoxicity of medicinal plants should be associated with other compounds, such as pyrrolizidine alkaloids [43].

**Table 3 plants-11-02114-t003:** Total protein content (*r*TP), catalytic activities, and concentrations of liver antioxidant enzymes in rats exposed to different doses of compounds found in *N. cataria* extracts and CCl_4_.

Group/Compounds	Conc. (mg/kg bwt)	*r*TP (g/L)	*r*SOD (U/mg)	*r*TBARS (nmol/mg)	*r*CAT (U/mg)	*r*GSH (mg/g)
Control group		^1^ 27.43 ± 0.21 ^†^	5.26 ± 0.14 ^†^	2.15 ± 0.32 ^†^	134.17 ± 0.15 ^†^	17.32 ± 0.20 ^†^
CCl_4_	1 mL/kg	25.43 ± 0.53 ***	2.53 ± 0.21 ***	10.78 ± 0.32 ***	107.56 ± 0.30 ***	6.42 ± 0.41 ***
quinic acid	50	22.43 ± 0.06 ***^†^	2.74 ± 0.12 ***	2.54 ± 0.21 ^†^	108.34 ± 0.32 ***^†^	8.04 ± 0.32 ***^†^
	100	31.45 ± 0.21 ***^†^	2.81 ± 0.32 ***	2.10 ± 0.43 ^†^	109.17 ± 0.51 ***^†^	8.43 ± 0.27 ***^†^
	200	21.32 ± 0.45 ***^†^	2.91 ± 0.26 ***	1.98 ± 0.32 ***^†^	110.56 ± 0.32 ***^†^	9.21 ± 0.32 ***^†^
protocatechuic acid	50	32.79 ± 0.46 ***^†^	2.66 ± 0.19 ***	2.95 ± 0.11 ^†^	110.25 ± 0.36 ***^†^	7.17 ± 0.63 ***^†^
	100	29.18 ± 0.95 ***^†^	2.69 ± 0.48 ***	3.18 ± 0.17 ***^†^	110.65 ± 0.91 ***^†^	7.34 ± 0.64 ***^†^
	200	30.05 ± 0.09 ***^†^	2.75 ± 0.14 ***	3.20 ± 0.47 ***^†^	113.89 ± 0.56 ***^†^	7.95 ± 0.46 ***^†^
chlorogenic acid	50	28.33 ± 0.47 ***^†^	2.88 ± 0.05 ***	2.43 ± 0.13 ^†^	110.11 ± 0.55 ***^†^	10.15 ± 0.28 ***^†^
	100	26.18 ± 0.19 ***^†^	2.95 ± 0.18 ***	2.37 ± 0.53 ^†^	112.28 ± 0.32 ***^†^	11.92 ± 0.31 ***^†^
	200	25.48 ± 0.11 ***	2.98 ± 0.33 ***	2.25 ± 0.46 ^†^	112.78 ± 0.28 ***^†^	12.28 ± 0.25 ***^†^
*p*-hydroxybenzoic acid	50	20.66 ± 0.16 ***^†^	2.59 ± 0.22 ***	3.99 ± 0.47 ***^†^	107.98 ± 0.79 ***	7.02 ± 0.17 ***^†^
	100	23.17 ± 0.96 ***^†^	2.63 ± 0.24 ***	3.90 ± 0.79 ***^†^	108.11 ± 0.36 ***^†^	7.49 ± 0.87 ***^†^
	200	23.11 ± 0.36 ***^†^	2.90 ± 0.33 ***	3.69 ± 0.09 ***^†^	108.92 ± 0.46 ***^†^	7.51 ± 0.42 ***^†^
caffeic acid	50	29.83 ± 0.45 ***^†^	2.64 ± 0.53 ***	4.18 ± 0.31 ***^†^	108.49 ± 0.25 ***^†^	6.81 ± 0.22 ***
	100	27.55 ± 0.36^†^	2.66 ± 0.34 ***	4.02 ± 0.07 ***^†^	110.71 ± 0.69 ***^†^	6.92 ± 0.13 ***
	200	27.19 ± 0.16^†^	2.72 ± 0.47 ***	3.97 ± 0.71 ***^†^	111.31 ± 0.33 ***^†^	7.14 ± 0.74 ***^†^
syringic acid	50	22.14 ± 0.32 ***^†^	2.78 ± 0.77 ***	2.23 ± 0.14 ^†^	115.28 ± 0.71 ***^†^	11.98 ± 0.49 ***^†^
	100	28.18 ± 0.23 ***^†^	3.18 ± 0.65 ***^†^	2.20 ± 0.38 ^†^	116.22 ± 0.24 ***^†^	12.92 ± 0.28 ***^†^
	200	24.65 ± 0.45 ***^†^	3.55 ± 0.39 ***^†^	2.19 ± 0.24 ^†^	118.39 ± 0.44 ***^†^	13.45 ± 0.36 ***^†^
*p*-coumaric acid	50	24.78 ± 0.15 ***^†^	2.74 ± 0.12 ***	2.72 ± 0.28 ^†^	112.28 ± 0.35 ***^†^	9.58 ± 0.27 ***^†^
	100	23.38 ± 0.30 ***^†^	2.79 ± 0.18 ***	2.58 ± 0.16 ^†^	112.99 ± 0.76 ***^†^	10.81 ± 0.14 ***^†^
	200	25.47 ± 0.12 ***	2.92 ± 0.74 ***	2.49 ± 0.71 ^†^	113.85 ± 0.29 ***^†^	11.17 ± 0.47 ***^†^
ferulic acid	50	28.91 ± 0.16 ***^†^	2.79 ± 0.61 ***	3.01 ± 0.19 ***^†^	113.11 ± 0.13 ***^†^	8.67 ± 0.37 ***^†^
	100	25.71 ± 0.63 ***	2.85 ± 0.37 ***	2.87 ± 0.22 ^†^	114.26 ± 0.34 ***^†^	8.99 ± 0.47 ***^†^
	200	23.33 ± 0.24 ***^†^	2.91 ± 0.11 ***	2.64 ± 0.17 ^†^	116.96 ± 0.26 ***^†^	9.25 ± 0.66 ***^†^
luteolin-7-*O*-glucoside	50	21.74 ± 0.29 ***^†^	2.85 ± 0.27 ***	2.98 ± 0.03 ^†^	109.45 ± 0.62 ***^†^	8.04 ± 0.72 ***^†^
	100	26.41 ± 0.39 ***^†^	2.99 ± 0.39 ***	2.90 ± 0.15 ^†^	110.21 ± 0.32 ***^†^	8.19 ± 0.37 ***^†^
	200	25.77 ± 0.07 ***	3.10 ± 0.09 ***^†^	2.61 ± 0.13 ^†^	114.65 ± 0.16 ***^†^	8.41 ± 0.11 ***^†^
quercetin-3-*O*-glucoside	50	^1^ 27.43 ± 0.21^†^	5.26 ± 0.14 ^†^	2.15 ± 0.32 ^†^	134.17 ± 0.15 ^†^	17.32 ± 0.20 ^†^
	100	25.43 ± 0.53 ***	2.53 ± 0.21 ***	10.78 ± 0.32 ***	107.56 ± 0.30 ***	6.42 ± 0.41 ***
	200	24.57 ± 0.34 ***^†^	2.59 ± 0.13 ***	3.11 ± 0.16 ***^†^	110.22 ± 0.28 ***^†^	7.99 ± 0.24 ***^†^
quercetin-3-*O*-rutinoside	50	22.18 ± 0.14 ***^†^	2.69 ± 0.08 ***	2.95 ± 0.39 ^†^	112.74 ± 0.36 ***^†^	8.10 ± 0.31 ***^†^
	100	25.96 ± 0.19 ***	2.74 ± 0.16 ***	2.78 ± 0.44 ^†^	115.18 ± 0.58 ***^†^	8.24 ± 0.77 ***^†^
	200	27.72 ± 0.11 ^†^	2.71 ± 0.19 ***	3.14 ± 0.26 ***^†^	116.78 ± 0.07 ***^†^	8.01 ± 0.64 ***^†^
rosmarinic acid	50	24.79 ± 0.46 ***^†^	2.75 ± 0.13 ***	3.08 ± 0.47 ***^†^	117.96 ± 0.45 ***^†^	8.09 ± 0.43 ***^†^
	100	25.11 ± 0.35 ***	2.98 ± 0.47 ***	2.97 ± 0.58 ^†^	120.47 ± 0.67 ***^†^	8.20 ± 0.27 ***^†^
	200	22.13 ± 0.45 ***^†^	2.57 ± 0.03 ***	2.65 ± 0.18 ^†^	109.52 ± 0.14 ***^†^	10.25 ± 0.07 ***^†^
quercitrin	50	21.18 ± 0.33 ***^†^	2.68 ± 0.09 ***	2.24 ± 0.08 ^†^	110.33 ± 0.23 ***^†^	11.78 ± 0.09 ***^†^
	100	24.79 ± 0.26 ***^†^	2.99 ± 0.12 ***	2.22 ± 0.09 ^†^	110.98 ± 0.32 ***^†^	11.26 ± 0.36 ***^†^
	200	21.53 ± 0.53 ***^†^	2.86 ± 0.54 ***	2.05 ± 0.74 ^†^	112.56 ± 0.31 ***^†^	12.43 ± 0.84 ***^†^
kaempferol-3-*O*-glucoside	50	28.43 ± 0.54 ***^†^	2.94 ± 0.23 ***	1.95 ± 0.51 ***^†^	114.52 ± 0.53 ***^†^	13.11 ± 0.31 ***^†^
	100	25.54 ± 0.11 ***	3.72 ± 0.77 ***^†^	1.64 ± 0.32 ***^†^	117.32 ± 0.21 ***^†^	13.32 ± 0.43 ***^†^
	200	21.54 ± 0.18 ***^†^	2.96 ± 0.16 ***	3.01 ± 0.22 ***^†^	111.27 ± 0.21 ***^†^	11.98 ± 0.66 ***^†^
quercetin	50	21.98 ± 0.47 ***^†^	3.14 ± 0.13 ***^†^	2.78 ± 0.61 ^†^	113.78 ± 0.36 ***^†^	13.56 ± 0.19 ***^†^
	100	26.74 ± 0.07 ***^†^	3.25 ± 0.34 ***^†^	2.47 ± 0.07 ^†^	119.14 ± 0.25 ***^†^	14.76 ± 0.11 ***^†^
	200	22.74 ± 0.11 ***^†^	2.97 ± 0.17 ***	2.01 ± 0.26 ^†^	113.78 ± 0.51 ***^†^	14.27 ± 0.09 ***^†^
luteolin	50	24.71 ± 0.24 ***^†^	3.15 ± 0.66 ***^†^	2.48 ± 0.41 ^†^	115.63 ± 0.14 ***^†^	14.98 ± 0.17 ***^†^
	100	25.98 ± 0.64 ***	3.67 ± 0.47 ***^†^	2.22 ± 0.45 ^†^	118.39 ± 0.33 ***^†^	15.36 ± 0.33 ***^†^
	200	24.69 ± 0.37 ***^†^	2.57 ± 0.57 ***	3.28 ± 0.49 ***^†^	108.49 ± 0.34 ***^†^	7.58 ± 0.31 ***^†^
kaempferol	50	23.15 ± 0.41 ***^†^	2.59 ± 0.64 ***	2.97 ± 0.34 ^†^	110.64 ± 0.97 ***^†^	8.25 ± 0.16 ***^†^
	100	22.14 ± 0.12 ***^†^	2.74 ± 0.08 ***	2.68 ± 0.09 ^†^	111.93 ± 0.13 ***^†^	10.92 ± 0.06 ***^†^
	200	28.14 ± 0.05 ***^†^	2.78 ± 0.16 ***	3.15 ± 0.19 ***^†^	115.97 ± 0.14 ***^†^	10.44 ± 0.22 ***^†^
isorhamnetin	50	26.81 ± 0.31 ***^†^	2.82 ± 0.64 ***	3.10 ± 0.24 ***^†^	119.36 ± 0.37 ***^†^	10.18 ± 0.20 ***^†^
	100	26.93 ± 0.12 ***^†^	2.95 ± 0.47 ***	2.99 ± 0.11 ^†^	120.55 ± 0.96 ***^†^	12.93 ± 0.13 ***^†^
	200	25.49 ± 0.17 ***	2.99 ± 0.13 ***	3.01 ± 0.17 ***^†^	122.68 ± 0.11 ***^†^	9.16 ± 0.41 ***^†^

^1^ Values represent mean ± SEM from three independent experiments; n = 5 rats per group; * *p* < 0.05 when compared with the negative control group; ^†^ *p* < 0.05 when compared with the CCl_4_ control group.

Quercitrin at 200 mg/kg bwt increased the catalytic activity of *r*SOD, decreased the concentration of *r*TBARS, increased the catalytic activity of *r*CAT, and increased the concentration of *r*GSH to the values 1.47-fold higher, 6.57-fold lower, 1.09-fold higher, and 2.07-fold higher than in the control group, respectively, matching the hepatoprotective profile of the extract itself. Related to quercitrin, chlorogenic acid and quinic acid at the highest concentration exerted slightly less protection against CCl_4_ in terms of *r*SOD (1.25-fold lower and 1.28-fold lower) and *r*CAT (1.04-fold lower and 1.06-fold lower) catalytic activities, as well as in terms of *r*GSH concentration (1.08-fold lower and 1.45-fold lower), at the same time allowing a higher concentration of *r*TBARS to be formed (1.37-fold higher and 1.21-fold higher). Related to the olive oil, all the found secondary metabolites decreased the catalytic activity of *r*CAT as the concentration applied increased (Table 3); this led to speculation that somehow new amounts of *r*H_2_O_2_ could have been produced in situ.

This expected outcome could occur upon the initial *r*H_2_O_2_ decomposition, catalyzed by *r*CAT during the “catalactic” reaction (Figure 2), and the formation of oxyferryl species with a porphyrin–radical cation, called the rat catalase compound I (*r*CATCmpd I), that may exist in either an oxoferryl porphyrin π-cation radical state (the Por·+-Fe^IV^ = O state, Figure 2, step 3) or a Por-Fe^IV^-OH state (Figure 2, step 4) [44,45,46,47,48,49,50,51,52,53,54]. Here, the noted secondary metabolites (FlaOHs) were likely the precursors for the induction of a higher amount of hydrogen peroxide: while interacting with *r*CAT, they stabilized the *r*CATCmpd I and enhanced the formation of *r*CATCmpd II, at the same time transforming themselves to the corresponding semiquinone radical form, *r*FlaO^•−^ (Figure 2, step 12), that initiated a chain of reactions (Figure 2, steps 13–15) to form new amounts of *r*H_2_O_2_ [55]. 

Recalling the pro-*r*H_2_O_2_ behavior of FlaOHs, to gather insights into the formation of new *r*H_2_O_2_ quantities at the molecular level, measured by the decrease in the catalytic activity of *r*CAT (Table 3), the secondary metabolites of *N. cataria* (Table 1, Figure 1) were subjected to a structure-based (SB) investigation within the *r*CATCmpd I (Figure 2, Appendix A, Appendix A) for elucidating the origin of *r*CATCmpd II. Hence, quercitrin at its putative bioactive conformation (Figure 2A) contributed by means of its A-ring’ C5-OH group, observed to be orthogonally related to the *r*CATCmpd I’s oxoferryl porphyrin π-cation radical (Figure 2A), in a position to establish a strong hydrogen bond (HB) (*d*_HB_ = 2.825 Å) and donate a proton and stabilize the *r*CATCmpd I. The formed quercitrinyl radical could lead to the conversion of molecular oxygen into *r*H_2_O_2_ (Figure 2, steps 13–15)_._ This particular molecular alignment could be a consequence of an electrostatic attraction between A-rings’ C7-OH portion and *r*Lys169, being that the A ring itself was stabilized by van der Waals interactions with *r*Leu165. Furthermore, both the C-ring’s C4-carbonyl group and the axially oriented L-rhamnose’s C2′-OH group established electrostatic interactions with *r*His75, hence making the first-level barrier for preventing the *r*H_2_O_2_ to enter the active site and initiate another “catalatic” reaction (Figure 2, step 1). The active site was “completely locked down” for *r*H_2_O_2_ upon the electrostatic interactions of L-rhamnose’s equatorial C3′-OH group with either *r*His75 or *r*Asn148, while the L-rhamnose’s equatorial C4′-OH portion additionally sealed the cavity while interacting correspondingly with *r*Asn148. The L-rhamnose’s alignment was supported by the favorable steric interactions of C5′-CH_3_ with *r*Phe153. The overall quercitrin binding mode was significantly conditioned by B-rings’ C3′-OH and C4′-OH groups’ negative interactions with *r*Asp128 and *r*Gln128, respectively. 

Finally, the quinic acid alone (Figure 2C) adopted a binding conformation nearby the *r*CATCmpd I, providing its stabilization by means of weak hydrogen bonding (*d*_HB_ = 3.313 Å) via C4-OH. The unsubstituted C5-OH portion electrostatically interacted with *r*His75, while C4-OH interacted similarly with *r*Asn128. As for chlorogenic acid, the C1 carboxyl group made electrostatic interactions with *r*Asp128, as well as the additional ones with *r*Gln128. The C1-OH portion provoked the induced dipole interactions with *r*Val128.

##### The Hepatocytes Cell Membrane Status

The catalytic activities of *r*AST, *r*ALT, *r*ALP, and *r*γ-GT may be considered sensitive indicators of acute liver damage, whereas the catalytic activities of *r*ALP and *r*γ-GT alone are thought to be markers of bile duct damage (Table 4 and Table 5) [53]. Herein, as expected, the CCl_4_-induced formation of *r*MDA (Table 2: group II) resulted in a significant increase in *r*AST and *r*ALT catalytic activities (Table 4: group II vs. group I, 22.51-fold and 3.06-fold increment, respectively), implying the hepatocyte’s membrane disruption and likely the hepatotoxicity [56]. Furthermore, the *N. cataria* FME as a stand-alone application reduced the catalytic activity of *r*AST referred to the control group level (Table 4: group III vs. group I) confirming that the extract has not induced any hepatotoxicity (Table 2). However, the slightly *r*ALT increased catalytic activity (Table 4: group III vs. group I, 1.82-fold increase) could be a marker of mild hepatotoxicity [57]. The 22.16-fold and 1.66-fold higher catalytic activities of *r*AST and *r*ALT associated with CCl_4_-treated samples also support the *N. cataria* FME lack of hepatotoxicity (Table 4: group II vs. group III).

**Table 4 plants-11-02114-t004:** Catalytic activities of serum biochemical markers within the rats exposed to different doses of *N. cataria* extracts and CCl_4_.

Group	*r*AST (U/L)	*r*ALT (U/L)	*r*ALP (U/L)	*r*γ-GT (U/L)
I	^1^ 3.16 ± 0.02	20.50 ± 0.23	266.27 ± 0.21	4.10 ± 0.03
II	71.12 ± 0.08 *	62.69 ± 0.30 *	359.18 ± 0.18 *	16.88 ± 0.15 *
III	3.21 ± 0.02 ^†^	37.75 ± 0.09 *^†^	269.44 ± 0.13 ^†^	4.18 ± 0.14 ^†^
IV	10.64 ± 0.25 *^†^	36.49 ± 0.12 *^†^	221.26 ± 0.19 *^†^	7.37 ± 0.06 *^†^
V	9.81 ± 0.09 *^†^	30.71 ± 0.02 *^†^	356.29 ± 0.25 *	8.44 ± 0.08 *^†^
VI	5.42 ± 0.01 *^†^	61.59 ± 0.23 *	297.82 ± 0.27 *^†^	10.81 ± 0.08 *^†^
VII	4.83 ± 0.04 ^†^	58.23 ± 0.18 *^†^	284.34 ± 0.12 *^†^	7.37 ± 0.31 *^†^
VIII	3.78 ± 0.09 ^†^	39.04 ± 0.13 *^†^	271.89 ± 0.14 *^†^	5.02 ± 0.24 ^†^
IX	3.60 ± 0.04 ^†^	69.52 ± 0.14 *^†^	349.44 ± 0.17 *	11.55 ± 0.11 *^†^
X	3.92 ± 0.03 ^†^	78.56 ± 0.28 *^†^	324.30 ± 0.13 *^†^	14.25 ± 0.05 *^†^
XI	11.13 ± 0.12 *^†^	39.98 ± 0.06 *^†^	246.85 ± 0.09 *^†^	28.00 ± 0.21 *^†^
XII	2.29 ± 0.01 ^†^	22.86 ± 0.11 ^†^	329.11 ± 0.31 *^†^	9.34 ± 0.09 *^†^
XIII	3.46 ± 0.03 ^†^	28.13 ± 0.09 *^†^	351.91 ± 0.36 *	29.48 ± 0.01 *^†^
XIV	11.87 ± 0.11 *^†^	46.95 ± 0.14 *^†^	456.12 ± 0.14 *^†^	17.20 ± 0.04 *

I, Control group, animals treated orally for five days with distilled water and then intraperitoneally (*i.p*.) injected with 1 mL/kg body weight (bwt) in olive oil; II, CCl_4_ 1 mL/kg *i.p*.; III, *N.cataria* FME 200 mg/kg; IV, *N.cataria* LME 200 mg/kg; V, *N.cataria* SME 200 mg/kg; VI, *N. cataria* FME 50 mg/kg+CCl_4_; VII, *N. cataria* FME 100 mg/kg+CCl_4_; VIII, *N. cataria* FME 200 mg/kg+CCl_4_; IX, *N. cataria* LME 50 mg/kg+CCl_4_; X, *N. cataria* LME 100 mg/kg+CCl_4_; XI, *N. cataria* LME 200 mg/kg+CCl_4_; XII, *N. cataria* SME 50 mg/kg+CCl_4_; XIII, *N. cataria* SME 100 mg/kg+CCl_4_; XIV, *N. cataria* SME 200 mg/kg+CCl_4_; ^1^ Values represent mean ± SEM from three independent experiments; n = 5 rats per group; * *p* < 0.05 when compared with the negative control group; ^†^ *p* < 0.05 when compared with the CCl_4_ control group. Results are presented as equivalents of total protein concentration.

Moreover, while counteracting the CCl_4_ effects, the extract decreased the catalytic activities of *r*AST and *r*ALT in a dose-dependent way (Table 4: groups VI–VIII), as the most concentrated application (200 mg/kg btw) almost re-established the full catalytic activity of *r*AST (Table 4: group VIII vs. group I), repairing the CCl_4_-caused damage (Table 4: group VIII vs. group II, 18.81-fold lower catalytic activity). As expected, the catalytic activity of *r*ALT was 1.90-fold higher within the untreated sample (Table 4: group VIII vs. group I) and 1.61-fold lower in the presence of the hazardous compound (Table 4: group II vs. group VIII). 

The catalytic activities of *r*ALP and *r*γ-GT were also concentration-dependently downgraded [58]. Hence, being increased alongside *r*AST, the catalytic activity of *r*ALP (Table 4: group II vs. group I, 1.35-fold increment) implied that the stand-alone administration of CCl_4_ could even cause hepatotoxicity [37], as validated by the increased catalytic activity of *r*γ-GT (Table 4: group II vs. group I, 4.12-fold, see further discussion). The *N. cataria* FME, however, proved to exert hepato-protective effects (Table 4: group III), as no hepatotoxicity was associated with the opposite performances of both *r*AST and *r*ALP (Table 4: group III) [37], in agreement with the changes of *r*γ-GT catalytic activity (Table 4: group III). At the FME highest concentration (200 mg/kg bwt) the CCl_4_ toxic effect was antagonized (Table 4: group VIII vs. group II, 1.32-fold drop in *r*ALP’s catalytic activity), and the inflammation calmed down through the re-establishment of bile activity (note the 3.36-fold lower catalytic activity of *r*γ-GT within Table 4: group VIII appertained to Table 4: group II).

Consistently, the pure metabolites of *N. cataria* FME (Table 1, Figure 1) unambiguously protected the hepatocytes’ membrane and the bile (Table 5). Thus, at the highest dosage, quercitrin completely suppressed the catalytic activity of *r*AST induced by CCl_4_ (Table 5: 16.99-fold lower catalytic activity) and calmed down *r*ALT, *r*ALP, and *r*γ-GT. Chlorogenic acid was slightly less efficient in protecting the liver from CCl_4_, leading to 1.04-fold higher, 1.04-fold lower, 1.02-fold higher, and 1.55-fold higher catalytic activities of *r*AST, *r*ALT, ALP, and *r*γ-GT than quercitrin. Furthermore, in agreement with the results of redox markers, quinic acid showed to be the weakest hepato-protector, displaying the least reduction of *r*AST, *r*ALT, *r*ALP, and *r*γ-GT catalytic activities, being 1.21-fold higher, 1.14-fold higher, 1.02-fold higher, and 1.91-fold higher than that of quercitrin.

#### 2.2.2. Hepatotoxic and Hepatoprotective Features of *N. cataria* Leaf Methanol Extract (LME) 

##### The Hepatocytes Redox Status

The *N. cataria* LME caused the catalytic activity of *r*SOD, the catalytic activity of *r*CAT, and the concentration of *r*GSH to be at 79.9%, 97.14%, and 83.04% of the base levels, respectively (Table 2: group IV vs. group I), whereas the concentration of *r*TBARS was 1.09-fold higher, not enough to be considered a marker of hepatotoxicity if compared with CCl_4_ (1.26-fold higher value, 4.40-fold lower value, 1.20-fold higher value, and 2.12-fold higher value, respectively; see Table 2: group IV vs. group II). Nevertheless, in a hepatoprotective fashion, *N. cataria* LME exerted plausible features in the concentration of 100 mg/kg bwt (the catalytic activity of *r*SOD, the concentration of *r*TBARS, the catalytic activity of *r*CAT, and the concentration of *r*GSH were 67.84%, 83.10%, 96.25%, and 82.67% of the control group values, see Table 2: group X vs. group I, and 1.49-fold higher, 5.77-fold lower, 1.19-fold higher, and 2.07-fold above the control value; see Table 2: group X vs. group II, respectively). Whereas when comparing the 100 mg/kg bwt *N. cataria* LME with 200 mg/kg bwt *N. cataria* FME, the leaf extract showed a lower efficacy to take under control the *r*SOD catalytic activity and *r*TBARS concentration but showed a better profile in terms of *r*CAT catalytic activity and *r*GSH concentration (Table 2: compare group X vs. group VIII). 

The *N. cataria* LME hepatoprotective effects could be mainly attributed to chlorogenic and rosmarinic acids (Table 3), which at 100 mg/kg bwt increased the *r*SOD catalytic activity concerning CCl_4_ (1.60-fold increase), partially neutralizing the damage in comparison to olive oil as a positive control (50.95% of the basal catalytic activity of *r*SOD restored). Regarding *r*TBARS, rosmarinic acid almost fully prevented their formation (only 1.04-fold higher concentration was measured related to the olive oil upon the compound’s administration against the hazardous chemical entity). However, rosmarinic acid alone was confirmed not as hepatoprotective by analyzing the catalytic activity of *r*CAT and the concentration of *r*GSH; the compound slightly increased the redox markers (1.03-fold and 1.83-fold, respectively), below basal values levels (82.23% and 68.01%, respectively), implying that further studies are required to determine whether the compounds could act in synergistic, antagonistic and/or additive ways.

Rosmarinic acid, being a caffeic acid analog, was found to adopt a binding mode similar to that of chlorogenic acid (Figure 2D) in which the C4-OH established a strong hydrogen bond (*d*_HB_ = 2.820 Å) with *r*CATCmpd I, while the dihydroxyphenyl-lactic acid’s *para* and *meta* OH moieties established electrostatic interactions with *r*Lys177 and *r*Asp128, respectively.

##### The Hepatocytes Cell Membrane Status

Besides the favorable impact on the redox status (Table 2), the potential therapeutic application of *N. cataria* LME has been questioned and at least remained elusive. Hence, being administered solely, among the three types of *N. cataria* extracts, LME caused the highest catalytic activity of *r*AST (3.31-fold higher than FME; compare groups IV and III within Table 4; and 1.08-fold higher than SME; compare groups IV and V within Table 4).

Furthermore, analyzing the catalytic activity of *r*ALT, it seems that the liver tolerated better LME as xenobiotic than FME (1.03-fold lower value, compare groups IV and III within Table 4) but showed a more pronounced reaction than upon the administration of SME (1.19-fold higher value, compare groups IV and V within Table 4). Nevertheless, the catalytic activities of both enzymes were increased upon *N. cataria* LME administration, possibly indicating hepatotoxicity [57]. However, the LME forced the lowest catalytic activity of *r*ALP (1.34-fold lower than caused by FME, compare groups IV and III within Table 4, and 1.61-fold lower than caused by SME; compare groups IV and V within Table 4), for which hepatotoxicity was excluded [37]. The opposite conclusion was made by analyzing the catalytic activity of *r*γ-GT (1.76-fold higher and 1.14-fold lower values against FME and SME, respectively), indicating the bile mild affection.

Regarding toxicity prevention, at 100 mg/kg bwt, LME seemed to neutralize the adverse effects of CCl_4_ associated through the re-established catalytic activities of *r*AST, *r*ALP, and *r*γ-GT (1.24-fold higher, 1.22-fold higher, and 3.48-fold higher than the control group values; compare Table 4: group X vs. group I, and 18.14-fold lower, 1.11-fold lower, 1.18-fold lower than within the CCl_4_-treated group values; Table 4: compare group X vs. group II, respectively). However, the same formulation failed to fix the catalytic activity of *r*ALT (the value 1.25-fold higher after CCl_4_ treatment; Table 4: compare group X vs. group II), revealing the CCl_4_-induced hepatotoxicity [57]. Further confirmation that *N. cataria* LME may not be an optimal phytotherapy preparation was obtained with the lack of consistency between the administered concentration and the impact on serum toxicity markers (Table 2 vs. Table 4). Hence, the concentration of 50 mg/kg bwt appeared to be more effective while restoring basal catalytic activities of *r*AST and *r*γ-GT, whereas the concentration of 200 mg/kg bwt was more beneficial while lowering the catalytic activities of *r*ALT and *r*ALP. Therefore, additional studies are likely required before claiming the *N. cataria* LME as safe as hepatoprotective supplements.

In this regard, a question was raised about the contribution of the rosmarinic acid (Table 5) in hepatocytes membrane protection, as well. The compound seemed to be protective against CCl_4_ in terms of *r*AST, *r*ALP, and *r*γ-GT catalytic activities (1.08-fold higher, 1.03-fold higher, and 2.00-fold higher than after treatment with olive oil; Table 5: compare group X vs. group I; and 15.76-fold lower, 1.32-fold lower, and 2.81-fold lower than values within the control group; Table 5: compare group X vs. group I, respectively), but only moderately protective seen through the catalytic activity of *r*ALT (1.64-fold higher and 1.84-fold lower than after treatment within olive oil or CCl_4_; compare Table 8: group X vs. group I, and Table 5: group X vs. group II, respectively), blending in the pattern expressed by the extract itself.

#### 2.2.3. Hepatotoxic and Hepatoprotective Features of *N. cataria* Stems Methanol Extract (SME)

##### The Hepatocytes’ Redox Status

The administration of *N. cataria* SME at the highest concentration raised the catalytic activity of *r*SOD to the highest level compared to the corresponding extracts from other plant organs (Table 2: group V, 1.25-fold lower *r*SOD catalytic activity related to reference group, i.e., Table 2: group I), causing likely no hepatocyte injury in comparison with the CCl_4_ effects (Table 2: group V, 1.76-fold higher *r*SOD catalytic activity compared to Table 2: group II). No significant harm to the liver was also confirmed upon analyzing the concentration of *r*TBARS (7.90-fold lower and 1.65-fold lower concentration of *r*TBARS concerning either olive oil or CCl_4_; see Table 2: group V vs. group I and Table 2: group V vs. group II, respectively), the catalytic activity of *r*CAT (1.17-fold higher and 1.05-fold lower concerning either olive oil or CCl_4_; see Table 2: group V vs. group I and Table 2: group V vs. group II, respectively), and the concentration of *r*GSH (1.08-fold lower and 2.31-fold higher related to either olive oil or CCl_4_; see Table 2: group V vs. group I and Table 2: group V vs. group II, respectively).

Furthermore, it should be stressed that the SME at a concentration of 50 mg/kg bwt caused the lowest decrease in *r*SOD catalytic activity and hence the best hepatoprotection against the radical-causing agent (Table 2, group XII vs. group II: catalytic activity of *r*SOD was 1.35-fold higher compared with the value caused by CCl_4_). A similar trend was spotted while discussing the *r*TBARS (Table 2, group XII vs. group II: concentration of *r*TBARS was 2.44-fold lower compared with the value caused by CCl_4_) and *r*GSH concentrations (Table 2, group XII vs. group II: concentration of *r*GSH was 2.00-fold lower compared with the value caused by CCl_4_), but not while considering the catalytic activity of *r*CAT (where extract in the concentration of 200 mg/kg bwt performed the best in terms of protecting liver form CCl_4_-based intoxication), revealing the limitation of the distinct formulation.

The *N. cataria* SME hepatocytes’ protection against CCl_4_ is likely due to chlorogenic acid (Table 1, Figure 1), as it can influence the redox markers (Table 3). At the concentration of 50 mg/kg bwt, chlorogenic acid increased the *r*SOD and *r*CAT catalytic activities (1.14-fold higher value and 1.02-fold higher value), increased the *r*GSH concentration (1.58-fold higher), and at the same time decreased the concentration of *r*TBARS (4.44-fold lower value). 

##### The Hepatocytes’ Cell Membrane Status

Notwithstanding the lack of *N. cataria* SME hepatotoxicity at the highest concentration, the extract caused some notable damage to the liver membrane and bile, as indicated by the catalytic activities of *r*AST, *r*ALT, *r*ALP, and *r*γ-GT (Table 4: group V vs. group I, 3.10-fold, 1.50-fold, 1.34-fold, and 2.06-fold higher values than within the control group, respectively); as for the catalytic activity of *r*ALP, the damage was even similar as after the CCl_4_ treatment (compare group V and group II in Table 4, 1.01-fold in the favor of CCl_4_), indicating, alongside with the catalytic activity of *r*AST, an appearance of mild hepatotoxicity [37] (the catalytic activities of *r*AST, *r*ALT, *r*γ-GT were 7.25-fold, 2.04-fold, and 2-fold lower within the group V of Table 4 than within the group II, respectively).

On the contrary, *N. cataria* SME applied antagonistically to CCl_4_ at a concentration of 50 mg/kg (Table 4: group XII) caused the highest hepatoprotection in agreement with the redox status analysis (Table 2). The catalytic activity of *r*AST was below the control group value (72.47% of the value within Table 4: group I) and was significantly lower than within the sample saturated with CCl_4_ (31.06-fold lower than within Table 4: group I), whereas the catalytic activity of *r*ALT was comparable to the control group value (compare group XII and group I of Table 4) but 2.72-fold lower than caused by CCl_4_ (Table 4: group XII vs. group II): no hepatotoxicity was observed. SE was less efficient in restoring the catalytic activities of *r*ALP and *r*γ-GT yet lowering them to the levels 1.09-fold and 1.81-fold below the ones caused by CCl_4_, respectively (see Table 4: group XII vs. group II), but no hepatotoxicity was observed, as well. Therefore, ad hoc formulated SE concentration could be safe for therapeutic administration.

Chlorogenic acid (Table 1, Figure 1) unambiguously neutralized the hepatocytes’ CCl_4_ harmfulness (Table 5), as it was not among the hepatotoxic compounds, causing the catalytic activities of *r*AST, *r*ALT, *r*ALP, and *r*γ-GT to be 1.04-fold higher, 1.04-fold lower, 1.02-fold higher, and 1.55-higher than CCl_4_, respectively. 

**Table 5 plants-11-02114-t005:** Catalytic activities of serum biochemical markers within the rats exposed to different doses of compounds found in *N. cataria* extracts and CCl_4_.

Group/Compounds	Conc. (mg/kg bwt)	*r*AST (U/L)	*r*ALT (U/L)	*r*ALP (U/L)	*r*γ-GT (U/L)
Control group		^1^ 4.08 ± 0.12 ^†^	21.37 ± 0.26 ^†^	272.94 ± 0.83 ^†^	4.01 ± 0.03 ^†^
CCl_4_	1 mL/kg	69.64 ± 0.41 *	64.72 ± 0.38 *	371.14 ± 0.46 *	15.78 ± 0.08 *
quinic acid	50	5.58 ± 0.08 *^†^	51.04 ± 0.62 *^†^	289.59 ± 0.33 *^†^	11.41 ± 0.03 *^†^
	100	5.12 ± 0.14 *^†^	45.94 ± 0.13 *^†^	285.17 ± 0.28 *^†^	9.23 ± 0.11 *^†^
	200	4.96 ± 0.07 ^†^	44.19 ± 0.27 *^†^	279.86 ± 0.19 ^†^	8.40 ± 0.26 *^†^
protocatechuic acid	50	9.11 ± 0.18 *^†^	59.67 ± 0.10 *^†^	323.16 ± 0.85 *^†^	12.53 ± 0.32 *^†^
	100	7.56 ± 0.06 *^†^	54.46 ± 0.34 *^†^	300.54 ± 0.62 *^†^	11.70 ± 0.10 *^†^
	200	7.01 ± 0.20 *^†^	51.74 ± 0.22 *^†^	298.44 ± 0.30 *^†^	9.29 ± 0.13 *^†^
chlorogenic acid	50	7.34 ± 0.05 *^†^	50.11 ± 0.41 *^†^	302.51 ± 0.49 *^†^	10.21 ± 0.15 *^†^
	100	6.03 ± 0.10 *^†^	42.67 ± 0.30 *^†^	284.53 ± 0.32 *^†^	9.28 ± 0.50 *^†^
	200	4.28 ± 0.07 ^†^	37.12 ± 0.28 *^†^	280.22 ± 0.41 *^†^	6.80 ± 0.77 *^†^
*p*-hydroxybenzoic acid	50	19.71 ± 0.08 *^†^	55.16 ± 0.52 *^†^	354.79 ± 0.97 *^†^	11.83 ± 0.65 *^†^
	100	14.28 ± 0.26 *^†^	52.11 ± 0.40 *^†^	331.60 ± 0.54 *^†^	10.91 ± 0.38 *^†^
	200	10.46 ± 0.37 *^†^	48.73 ± 0.13 *^†^	310.93 ± 0.19 *^†^	9.04 ± 0.50 *^†^
caffeic acid	50	27.53 ± 0.41 *^†^	63.61 ± 0.55 *^†^	360.38 ± 0.59 *^†^	13.91 ± 0.05 *^†^
	100	20.14 ± 0.52 *^†^	62.11 ± 0.43 *^†^	322.62 ± 0.74 *^†^	10.16 ± 0.37 *^†^
	200	19.97 ± 0.77 *^†^	48.23 ± 0.71 *^†^	307.08 ± 0.62 *^†^	9.10 ± 0.52 *^†^
syringic acid	50	16.46 ± 0.06 *^†^	62.39 ± 0.30 *^†^	382.63 ± 0.95 *^†^	13.67 ± 0.08 *^†^
	100	15.17 ± 0.17 *^†^	53.65 ± 0.44 *^†^	366.19 ± 0.48 *^†^	11.49 ± 0.23 *^†^
	200	12.39 ± 0.53 *^†^	41.25 ± 0.31 *^†^	350.02 ± 0.36 *^†^	10.42 ± 0.64 *^†^
*p*-coumaric acid	50	24.73 ± 0.46 *^†^	60.18 ± 0.68 *^†^	314.09 ± 0.43 *^†^	11.44 ± 0.13 *^†^
	100	20.16 ± 0.38 *^†^	54.03 ± 0.34 *^†^	296.91 ± 0.84 *^†^	9.97 ± 0.08 *^†^
	200	15.69 ± 0.18 *^†^	40.82 ± 0.08 *^†^	289.92 ± 0.38 *^†^	8.05 ± 0.61 *^†^
ferulic acid	50	8.28 ± 0.61 *^†^	52.10 ± 0.24 *^†^	312.14 ± 0.92 *^†^	11.77 ± 0.19 *^†^
	100	7.39 ± 0.05 *^†^	47.22 ± 0.56 *^†^	297.96 ± 0.43 *^†^	9.26 ± 0.46 *^†^
	200	5.94 ± 0.23 *^†^	41.69 ± 0.10 *^†^	288.71 ± 0.55 *^†^	8.10 ± 0.49 *^†^
luteolin-7-*O*-glucoside	50	23.17 ± 0.46 *^†^	59.71 ± 0.26 *^†^	319.37 ± 0.53 *^†^	13.78 ± 0.10 *^†^
	100	21.55 ± 0.83 *^†^	55.96 ± 0.09 *^†^	302.74 ± 0.33 *^†^	10.62 ± 0.41 *^†^
	200	17.49 ± 0.09 *^†^	39.28 ± 0.17 *^†^	281.93 ± 0.42 *^†^	9.11 ± 0.72 *^†^
quercetin-3-*O*-glucoside	50	7.82 ± 0.36 *^†^	51.69 ± 0.44 *^†^	304.03 ± 0.20 *^†^	10.58 ± 0.04 *^†^
	100	6.99 ± 0.42 *^†^	46.02 ± 0.13 *^†^	296.37 ± 0.17 *^†^	9.35 ± 0.50 *^†^
	200	5.17 ± 0.07 *^†^	40.33 ± 0.76 *^†^	284.46 ± 0.66 *^†^	6.43 ± 0.21 *^†^
quercetin-3-*O*-rutinoside	50	9.26 ± 0.18 *^†^	53.10 ± 0.22 *^†^	300.23 ± 0.44 *^†^	12.77 ± 0.16 *^†^
	100	7.33 ± 0.17 *^†^	48.47 ± 0.07 *^†^	294.16 ± 0.62 *^†^	9.16 ± 0.41 *^†^
	200	5.70 ± 0.30 *^†^	43.64 ± 0.39 *^†^	280.15 ± 0.38 *^†^	7.73 ± 0.06 *^†^
rosmarinic acid	50	5.71 ± 0.18 *^†^	43.01 ± 0.30 *^†^	297.55 ± 0.14 *^†^	10.40 ± 0.31 *^†^
	100	4.42 ± 0.20 ^†^	35.12 ± 0.22 *^†^	281.96 ± 0.35 *^†^	8.04 ± 0.13 *^†^
	200	4.13 ± 0.02 ^†^	32.64 ± 0.40 *^†^	273.58 ± 0.31 ^†^	5.61 ± 0.23 *^†^
quercitrin	50	4.72 ± 0.31 ^†^	47.66 ± 0.38 *^†^	299.48 ± 0.21 *^†^	9.56 ± 0.29 *^†^
	100	4.25 ± 0.10 ^†^	40.19 ± 0.09 *^†^	293.13 ± 0.84 *^†^	6.28 ± 0.08 *^†^
	200	4.10 ± 0.27 ^†^	38.62 ± 0.67 *^†^	274.22 ± 0.34 ^†^	4.40 ± 0.40 ^†^
quercetin-3-*O*-glucoside	50	7.82 ± 0.36 *^†^	51.69 ± 0.44 *^†^	304.03 ± 0.20 *^†^	10.58 ± 0.04 *^†^
	100	6.99 ± 0.42 *^†^	46.02 ± 0.13 *^†^	296.37 ± 0.17 *^†^	9.35 ± 0.50 *^†^
	200	5.17 ± 0.07 *^†^	40.33 ± 0.76 *^†^	284.46 ± 0.66 *^†^	6.43 ± 0.21 *^†^
kaempferol-3-*O*-glucoside	50	15.82 ± 0.09 *^†^	61.22 ± 0.08 *^†^	313.08 ± 0.54 *^†^	11.60 ± 0.08 *^†^
	100	13.95 ± 0.20 *^†^	53.18 ± 0.16 *^†^	302.60 ± 0.97 *^†^	10.35 ± 0.40 *^†^
	200	10.61 ± 0.15 *^†^	50.29 ± 0.23 *^†^	290.53 ± 0.72 *^†^	9.94 ± 0.31 *^†^
quercetin	50	6.28 ± 0.17 *^†^	50.52 ± 0.24 *^†^	317.37 ± 0.55 *^†^	11.64 ± 0.30 *^†^
	100	5.14 ± 0.62 *^†^	45.26 ± 0.61 *^†^	300.56 ± 0.31 *^†^	9.92 ± 0.07 *^†^
	200	4.49 ± 0.43 ^†^	41.40 ± 0.38 *^†^	291.74 ± 0.97 *^†^	7.50 ± 0.46 *^†^
luteolin	50	25.18 ± 0.63 *^†^	62.83 ± 0.49 *^†^	325.74 ± 0.30 *^†^	12.75 ± 0.32 *^†^
	100	22.79 ± 0.08 *^†^	57.01 ± 0.37 *^†^	299.41 ± 0.67 *^†^	10.06 ± 0.42 *^†^
	200	18.95 ± 0.84 *^†^	49.46 ± 0.26 *^†^	292.16 ± 0.11 *^†^	9.14 ± 0.58 *^†^
kaempferol	50	15.82 ± 0.09 *^†^	61.22 ± 0.08 *^†^	313.08 ± 0.54 *^†^	11.60 ± 0.08 *^†^
	100	13.95 ± 0.20 *^†^	53.18 ± 0.16 *^†^	302.60 ± 0.97 *^†^	10.35 ± 0.40 *^†^
	200	10.61 ± 0.15 *^†^	50.29 ± 0.23 *^†^	290.53 ± 0.72 *^†^	9.94 ± 0.31 *^†^
isorhamnetin	50	14.74 ± 0.44 *^†^	59.25 ± 0.67 *^†^	294.30 ± 0.26 *^†^	12.01 ± 0.18 *^†^
	100	11.62 ± 0.19 *^†^	52.11 ± 0.91 *^†^	291.62 ± 0.45 *^†^	9.93 ± 0.06 *^†^
	200	7.30 ± 0.07 *^†^	44.08 ± 0.33 *^†^	282.06 ± 0.36 *^†^	8.01 ± 0.32 *^†^

^1^ Values represent mean ± SEM from three independent experiments; n = 5 rats per group; * *p* < 0.05 when compared with the negative control group; ^†^ *p* < 0.05 when compared with the CCl_4_ control group.

### 2.3. N. cataria Extracts Impact the Genome In Vivo 

Once produced, the *r*CCl_3_^•^ to *r*CCl_3_OO^•^ radicals could reach the DNA and influence the *r*O_2_^•−^ that could attack *r*DNA nucleic bases or deoxyribose residues (Figure 1), producing damaged bases or strand disruption [29], cellular events that can be quantified by comet assay [58]. Nevertheless, a likely scenario is that *r*CCl_3_OO^•^ radical induces the formation of *r*MDA (Table 2, Table 3 and Table 4) that could interact with DNA to form adducts such as *r*M_1_G, *r*M_1_A, and *r*M_1_C (Figure 1). Some of the adducts might induce base transversions and transitions and the consequent ring-opening to form *rN*^2^-oxopropenyl-dG, *rN*^2^-oxopropenyl-dA or *rN*^2^-oxopropenyl-dC adducts that could lead to *r*DNA–*r*DNA inter-strand cross-linking or *r*DNA–protein inter-strand crosslinks [28], cellular events that can be evaluated by comet assay. Moreover, as the interaction of secondary metabolites with *r*CAT (Table 3, Figure 2, Figure 2) induced the formation of a new amount of hydrogen peroxide, the product’s degradation to a hydroxyl radical may endow in targeting the ribose within the *r*DNA strand with distinct reactive oxygen species, for which additional quantities of *r*MDA could emerge [28,59]. Therefore, based on *r*DNA *r*MDA-induced products, the effectiveness of FME, LME, and SME to either harm or protect *r*DNA after treatment with CCl_4_ can be evaluated, in which this aspect is herein discussed.

#### 2.3.1. Genotoxic and Antigenotoxic Activities of *N. cataria* Flower Methanol Extract (FME)

The CCl_4_ administration caused severe hepatocellular injury associated with the above-described cellular events, i.e., it significantly increased (*p* < 0.05) the level of *r*DNA damage (Table 6 and Table 7: group II) compared to the control group (Table 6 and Table 7: group I): 11.47-fold higher *r*tail moment value, 3.51-increment in the % *r*DNA in tail, and 9.48-fold increase in *r*tail length. In comparison to the CCl_4_ induced damage, the administration of *N. cataria* FME at 200 mg/kg bwt concentration caused no genotoxicity, leading to 10.07-fold lowering of *r*tail moment value, 2.90-fold decrease in the % *r*DNA in tail, and 8.50-fold lessen in *r*tail length (Table 6: group II vs. group III). However, related to the control group, some raising in the *r*tail moment (1.14-fold higher value), % *r*DNA in the tail (1.21-fold higher value), and *r*tail length (1.12-fold higher value) were also observed (Table 6: group III vs. group I), although with no significant differences (*p* > 0.05), which could be attributed to in situ generated *r*H_2_O_2_ (Figure 2) and the consequent formation of *r*MDA-mediated lesions.

**Table 6 plants-11-02114-t006:** DNA damage effect of *N. cataria* extracts on liver cells of albino Wistar rats.

Groups	*r*tail Moment	% *r*DNA in Tail	*r*tail Length
I	1.44 ± 0.32 ^a†^	3.08 ± 0.40 ^†^	2.60 ± 0.73 ^†^
II	16.51 ± 1.23 *	10.80 ± 1.50 *	24.65 ± 1.03 *
III	1.64 ± 0.63 ^†^	3.72 ± 0.94 ^†^	2.90 ± 0.86 ^†^
IV	2.99 ± 0.6 *^†^	4.22 ± 0.54 *^†^	3.17 ± 0.63 ^†^
V	4.72 ± 0.3 *^†^	7.62 ± 0.34 *^†^	5.6 ± 0.82 *^†^

^a^ Data are presented as the means ± SEM obtained from three independent experiments; n = 25 rats; 5 rats per group. I, Control group, animals treated orally for five days with distilled water and then intraperitoneally (*i.p*.) injected with 1 mL/kg body weight (bwt) in olive oil; II, CCl_4_ 1 mL/kg *i.p*.; III, *N. cataria* FME 200 mg/kg;, IV, *N. cataria* LME 200 mg/kg; V, *N. cataria* SME 200 mg/kg; * *p* < 0.05 when compared with the negative control group; ^†^ *p* < 0.05 when compared with the positive control group.

However, physiological intoxication with *r*H_2_O_2_ seemed to be irrelevant to the FME antigenotoxic features (Table 7). Thus, while an increase in total score was detected in rats exposed only to CCl_4_ (4.93-fold increment, Table 7: group II vs. group I), the levels of liver *r*DNA damage analyzed from the groups treated with different concentrations of *N. cataria* FME were significantly reduced (Table 7, groups VI-XIV). The extract at the highest concentration (Table 7: group VIII) reduced the CCl_4_, which caused the highest *r*DNA (2.24-fold lower total score than the hazardous agent, Table 7: group II vs. group VIII, but 1.83-fold higher one than within the control group, Table 7: group VII vs. group I), with a percentage reduction level of 73.13% and the absence of comet classes *3* and *4*. The finding that the *N. cataria* FME at the highest concentration was an efficient hepatoprotective supplement that caused no genotoxicity as well as supplementation raised an interesting question: “are the hepatoprotective extracts also antigenotoxic agents?” 

Similar to the extract, the secondary metabolites listed in Table 1 were also investigated for their assessment for antigenotoxic features to correlate them with extracts’ corresponding effects (Table 8). Quercitrin, chlorogenic acid, and quinic acid as representants of FME were found to exert antigenotoxic activity, associated with the absence of comet classes *3* and *4.* Compared to CCl_4_, quercitrin at 200 mg/kg bwt concentration severely reduced the CCl_4_, causing *r*DNA damage (causing a 2.35-fold lower total score than the hazardous agent, but 1.89-fold higher one than within the control group), with a percentage reduction of 74.08%. Furthermore, in the same concentration, chlorogenic acid reduced the total score in the damaged sample by 1.73-fold, still keeping it on a level 2.56-fold higher than within the undamaged one, and it was associated with a percentage reduction of 54.68%. Lastly, the quinic acid was efficient in protecting the liver from CCl_4_, with the percentage of reduction equal to 69.98, the 2.18-fold lower total score value than within the sample enriched CCl_4_ and 2.03-fold higher than within the untreated sample. 

**Table 7 plants-11-02114-t007:** DNA damage in livers of rats exposed to CCl_4_ and different doses of *N. cataria* extracts.

Groups	Comet class					Total Score ^1^	% R
	0	1	2	3	4		
I	73.90 ± 1.02	26.10 ± 0.82	0.00 ± 0.00	0.00 ± 0.00	0.00 ± 0.00	26.10 ± 0.82 ^†^	/
II	19.33 ± 0.34	59.70 ± 1.02	16.13 ± 0.30	4.84 ± 1.70	0.00 ± 0.00	106.50 ± 1.04 *	/
VI	38.60 ± 0.33	59.10 ± 1.40	2.30 ± 0.73	0.00 ± 0.00	0.00 ± 0.00	63.70 ± 0.34 *^†^	53.23
VII	44.86 ± 0.23	53.10 ± 0.20	2.04 ± 0.03	0.00 ± 0.00	0.00 ± 0.00	57.20 ± 0.30 *^†^	61.32
VIII	53.85 ± 0.40	44.61 ± 0.32	1.54 ± 0.80	0.00 ± 0.00	0.00 ± 0.00	47.70 ± 1.54 *^†^	73.13
IX	48.00 ± 0.23	46.00 ± 0.64	6.00 ± 0.50	0.00 ± 0.00	0.00 ± 0.00	58.00 ± 1.70 *^†^	60.32
X	49.95 ± 1.02	45.70 ± 0.54	4.35 ± 1.60	0.00 ± 0.00	0.00 ± 0.00	54.40 ± 1.02 *^†^	64.80
XI	40.00 ± 0.80	60.00 ± 0.30	0.00 ± 0.00	0.00 ± 0.00	0.00 ± 0.00	60.00 ± 0.12 *^†^	57.80
XII	62.20 ± 0.20	32.40 ± 0.92	4.05 ± 0.90	1.35 ± 0.52	0.00 ± 0.00	44.55 ± 0.54 *^†^	77.05
XIII	43.80 ± 0.82	47.90 ± 0.32	8.30 ± 0.54	0.00 ± 0.00	0.00 ± 0.00	64.50 ± 0.81 *^†^	52.24
XIV	47.90 ± 0.41	37.50 ± 0.90	12.50 ± 0.32	2.10 ± 0.30	0.00 ± 0.00	68.80 ± 0.46 *^†^	46.90

I, Control group, animals treated orally for five days with distilled water and then intraperitoneally (*i.p*.) injected with 1 mL/kg body weight (bwt) in olive oil; II, CCl_4_ 1 mL/kg *i.p*.; III, *N.cataria* FME 200 mg/kg; IV, *N.cataria* LME 200 mg/kg; V, *N.cataria* SME 200 mg/kg; VI, *N. cataria* FME 50 mg/kg+CCl_4_; VII, *N. cataria* FME 100 mg/kg+CCl_4_; VIII, *N. cataria* FME 200 mg/kg+CCl_4_; IX, *N. cataria* LME 50 mg/kg+CCl_4_; X, *N. cataria* LME 100 mg/kg+CCl_4_; XI, *N. cataria* LME 200 mg/kg+CCl_4_; XII, *N. cataria* SME 50 mg/kg+CCl_4_; XIII, *N. cataria* SME 100 mg/kg+CCl_4_; XIV, *N. cataria* SME 200 mg/kg+CCl_4_; ^1^ Values represent mean ± SEM from three independent experiments; n = 5 rats per group; * *p* < 0.05 when compared with the negative control group; ^†^ *p* < 0.05 when compared with the CCl_4_ control group. Results are presented as equivalents of total protein concentration.

**Table 8 plants-11-02114-t008:** DNA damage in livers of rats exposed to CCl_4_ and different doses of compounds found in *N. cataria* extracts.

Group/Compounds	Conc. (mg/kg bwt)	Comet Class					Total Score ^1^	% R
		0	1	2	3	4		
Control group		75.59 ± 1.30	24.41 ± 0.52	0.00 ± 0.00	0.00 ± 0.00	0.00 ± 0.00	24.41 ± 0.52 ^†^	/
CCl_4_	1 mL/kg	19.09 ± 0.57	58.57 ± 0.44	17.42 ± 0.29	4.92 ± 0.69	0.00 ± 0.00	108.17 ± 0.31 *	/
quinic acid	50	35.90 ± 0.67	60.08 ± 1.03	4.02 ± 0.50	0.00 ± 0.00	0.00 ± 0.00	68.12 ± 0.81 *^†^	47.82
	100	46.48 ± 0.93	50.67 ± 0.82	2.85 ± 0.43	0.00 ± 0.00	0.00 ± 0.00	56.37 ± 0.94 *^†^	61.84
	200	52.70 ± 0.03	45.05 ± 0.14	2.25 ± 0.52	0.00 ± 0.00	0.00 ± 0.00	49.55 ± 0.61 *^†^	69.99
protocatechuic acid	50	26.95 ± 1.01	58.36 ± 0.29	11.14 ± 0.37	3.55 ± 0.18	0.00 ± 0.00	91.29 ± 0.28 *^†^	20.15
	100	36.10 ± 0.57	47.56 ± 0.68	12.20 ± 0.66	4.14 ± 1.06	0.00 ± 0.00	84.38 ± 1.09 *^†^	28.40
	200	42.30 ± 0.22	38.70 ± 0.54	15.27 ± 0.58	3.73 ± 0.27	0.00 ± 0.00	80.43 ± 1.14 *^†^	33.12
chlorogenic acid	50	46.10 ± 0.91	48.65 ± 0.30	5.25 ± 0.37	0.00 ± 0.00	0.00 ± 0.00	59.15 ± 0.84 *^†^	58.52
	100	45.67 ± 0.69	48.20 ± 0.48	6.13 ± 0.79	0.00 ± 0.00	0.00 ± 0.00	60.46 ± 0.37 *^†^	56.96
	200	48.25 ± 0.63	41.13 ± 0.54	10.62 ± 0.40	0.00 ± 0.00	0.00 ± 0.00	62.37 ± 1.15 *^†^	54.68
*p*-hydroxybenzoic acid	50	31.82 ± 0.59	51.81 ± 0.33	13.98 ± 0.52	2.39 ± 0.38	0.00 ± 0.00	86.94 ± 0.33 *^†^	25.35
	100	33.48 ± 0.50	55.00 ± 1.42	6.85 ± 0.02	4.67 ± 0.50	0.00 ± 0.00	82.71 ± 0.84 *^†^	30.40
	200	29.09 ± 1.06	64.23 ± 1.11	4.63 ± 0.83	2.05 ± 0.44	0.00 ± 0.00	79.64 ± 1.25 *^†^	34.06
caffeic acid	50	44.42 ± 0.97	46.29 ± 0.24	9.29 ± 0.62	0.00 ± 0.00	0.00 ± 0.00	64.87 ± 0.34 *^†^	51.70
	100	46.80 ± 0.39	46.69 ± 0.52	6.51 ± 0.94	0.00 ± 0.00	0.00 ± 0.00	59.71 ± 0.99 *^†^	57.86
	200	49.12 ± 0.94	44.33 ± 0.56	5.49 ± 0.26	1.06 ± 0.31	0.00 ± 0.00	58.49 ± 0.26 *^†^	59.31
syringic acid	50	39.40 ± 0.23	43.76 ± 0.17	12.88 ± 0.19	3.96 ± 0.00	0.00 ± 0.00	81.40 ± 1.48 *^†^	31.96
	100	37.44 ± 0.33	50.05 ± 0.59	11.46 ± 0.27	1.05 ± 0.00	0.00 ± 0.00	76.12 ± 1.02 *^†^	38.26
	200	33.18 ± 0.61	59.61 ± 1.28	7.21 ± 0.03	0.00 ± 0.00	0.00 ± 0.00	74.03 ± 0.95 *^†^	40.76
*p*-coumaric acid	50	31.49 ± 0.52	61.47 ± 0.55	5.40 ± 0.68	1.64 ± 0.24	0.00 ± 0.00	77.19 ± 0.54 *^†^	36.99
	100	39.57 ± 0.49	51.01 ± 0.91	6.79 ± 0.32	2.63 ± 0.81	0.00 ± 0.00	72.48 ± 0.29 *^†^	42.61
	200	29.62 ± 0.22	69.10 ± 0.68	1.28 ± 0.08	0.00 ± 0.00	0.00 ± 0.00	71.66 ± 0.33 *^†^	43.59
ferulic acid	50	30.65 ± 0.34	59.23 ± 0.67	10.12 ± 0.49	0.00 ± 0.00	0.00 ± 0.00	79.47 ± 0.48 *^†^	34.26
	100	29.32 ± 0.50	64.20 ± 1.23	6.48 ± 0.33	0.00 ± 0.00	0.00 ± 0.00	77.16 ± 0.97 *^†^	37.02
	200	35.89 ± 1.02	58.14 ± 1.40	5.97 ± 0.25	0.00 ± 0.00	0.00 ± 0.00	70.08 ± 0.90 *^†^	45.48
luteolin-7-*O*-glucoside	50	29.96 ± 0.46	63.83 ± 0.30	6.21 ± 0.96	0.00 ± 0.00	0.00 ± 0.00	76.25 ± 0.82 *^†^	38.11
	100	32.59 ± 0.81	60.43 ± 0.26	6.98 ± 0.20	0.00 ± 0.00	0.00 ± 0.00	74.39 ± 0.39 *^†^	40.33
	200	31.62 ± 1.22	63.61 ± 1.00	4.77 ± 0.19	0.00 ± 0.00	0.00 ± 0.00	73.15 ± 0.08 *^†^	41.81
quercetin-3-*O*-glucoside	50	41.76 ± 0.16	50.32 ± 0.90	7.92 ± 0.05	0.00 ± 0.00	0.00 ± 0.00	66.16 ± 0.67 *^†^	50.16
	100	44.55 ± 1.06	51.77 ± 0.38	3.68 ± 0.40	0.00 ± 0.00	0.00 ± 0.00	59.13 ± 0.47 *^†^	58.55
	200	56.77 ± 1.31	40.77 ± 1.23	2.46 ± 0.14	0.00 ± 0.00	0.00 ± 0.00	45.69 ± 0.49 *^†^	74.59
quercetin-3-*O*-rutinoside	50	47.52 ± 0.96	35.05 ± 1.14	15.28 ± 1.31	2.15 ± 0.01	0.00 ± 0.00	72.06 ± 0.16 *^†^	43.11
	100	43.51 ± 0.59	44.64 ± 1.30	9.57 ± 0.66	2.28 ± 0.61	0.00 ± 0.00	70.62 ± 0.56 *^†^	44.83
	200	54.61 ± 0.16	32.64 ± 0.27	11.69 ± 0.35	1.06 ± 0.09	0.00 ± 0.00	59.20 ± 0.22 *^†^	58.46
rosmarinic acid	50	53.66 ± 1.12	38.05 ± 0.93	8.29 ± 0.81	0.00 ± 0.00	0.00 ± 0.00	54.63 ± 1.41 *^†^	63.92
	100	48.16 ± 0.82	51.84 ± 0.55	0.00 ± 0.00	0.00 ± 0.00	0.00 ± 0.00	51.84 ± 0.55 *^†^	67.25
	200	53.71 ± 0.35	42.44 ± 0.19	3.85 ± 0.47	0.00 ± 0.00	0.00 ± 0.00	50.14 ± 0.68 *^†^	69.28
quercitrin	50	39.30 ± 0.70	58.56 ± 1.05	2.14 ± 0.64	0.00 ± 0.00	0.00 ± 0.00	62.84 ± 0.46 *^†^	54.12
	100	45.25 ± 0.24	50.76 ± 0.16	1.99 ± 0.11	0.00 ± 0.00	0.00 ± 0.00	54.74 ± 0.07 *^†^	63.79
	200	55.00 ± 0.61	43.88 ± 0.78	1.12 ± 0.45	0.00 ± 0.00	0.00 ± 0.00	46.12 ± 0.98 *^†^	74.08
kaempferol-3-*O*-glucoside	50	39.77 ± 0.33	50.75 ± 1.52	9.48 ± 0.22	0.00 ± 0.00	0.00 ± 0.00	69.71 ± 0.62 *^†^	45.92
	100	39.52 ± 0.84	57.27 ± 1.03	1.96 ± 0.40	1.25 ± 0.03	0.00 ± 0.00	64.94 ± 1.25 *^†^	51.61
	200	56.90 ± 0.19	35.15 ± 0.65	7.95 ± 0.16	0.00 ± 0.00	0.00 ± 0.00	51.05 ± 0.51 *^†^	68.19
quercetin	50	51.28 ± 1.20	35.47 ± 0.37	10.62 ± 1.46	2.63 ± 0.16	0.00 ± 0.00	64.60 ± 0.68 *^†^	52.02
	100	57.02 ± 0.62	27.99 ± 0.60	14.99 ± 0.24	0.00 ± 0.00	0.00 ± 0.00	57.97 ± 0.66 *^†^	59.93
	200	63.54 ± 1.48	25.47 ± 0.61	9.40 ± 0.33	1.59 ± 0.08	0.00 ± 0.00	49.04 ± 0.42 *^†^	70.59
luteolin	50	36.39 ± 0.53	55.09 ± 0.95	6.49 ± 0.37	2.03 ± 0.08	0.00 ± 0.00	74.16 ± 0.60 *^†^	40.60
	100	52.45 ± 0.42	37.14 ± 0.22	9.26 ± 0.14	1.15 ± 0.12	0.00 ± 0.00	59.11 ± 0.93 *^†^	58.57
	200	58.75 ± 1.05	30.54 ± 0.80	10.71 ± 1.07	0.00 ± 0.00	0.00 ± 0.00	51.96 ± 0.30 *^†^	67.11
kaempferol	50	39.43 ± 1.15	55.12 ± 0.64	5.45 ± 0.16	0.00 ± 0.00	0.00 ± 0.00	66.02 ± 0.92 *^†^	50.32
	100	50.67 ± 0.50	39.23 ± 0.99	10.10 ± 0.94	0.00 ± 0.00	0.00 ± 0.00	59.43 ± 0.83 *^†^	58.19
	200	55.50 ± 0.72	31.32 ± 0.80	13.18 ± 0.37	0.00 ± 0.00	0.00 ± 0.00	57.68 ± 0.29 *^†^	60.28
isorhamnetin	50	33.93 ± 0.33	56.04 ± 1.20	10.03 ± 0.93	0.00 ± 0.00	0.00 ± 0.00	76.10 ± 0.31 *^†^	38.29
	100	55.12 ± 0.59	25.52 ± 0.60	16.28 ± 1.17	3.08 ± 0.13	0.00 ± 0.00	67.32 ± 0.10 *^†^	48.77
	200	58.30 ± 0.28	30.18 ± 0.56	11.52 ± 1.09	0.00 ± 0.00	0.00 ± 0.00	53.22 ± 0.87 *^†^	65.60

^1^ Values represent mean ± SEM from three independent experiments; n = 5 rats per group; * *p* < 0.05 when compared with the control group; ^†^ *p* < 0.05 when compared with the CCl_4_ group.

The antigenotoxic agents’ activity was further investigated through structure-based studies using *R. norvegicus* Topoisomerase IIα (*r*TopIIα), an enzyme that catalyzes in vivo the decatenation of damaged *r*DNA, a process quantified by the appearance of comets (Table 8) [60]. Hence, the *N. cataria* secondary metabolites bioactive conformations within the *r*DNA binding and cleavage domain of *r*TopIIα were compared against *r*MDA, in which the binding pose was calculated nearby four targetable nucleotides positioned within the sense strand, two consecutive guanines residues (*r*G41 and *r*G42) followed by a thymine (*r*T43) and an adenine (*r*A44), as well as in the proximity of two cytosine residues (*r*C11 and *r*C12) and one *r*G13 belonging to the complementary antisense strand (Figure 3, Appendix A), creating a favorable environment for the generation of *r*dG, *r*dA, and *r*dC and for triggering the related cellular events.

Chlorogenic acid likewise protected all the nucleotides (Figure 3B), where the *m*-OH portion of the caffeic acid formed HB with the C6-carbonyl group of the *r*G13 (*d*_HB_ = 3.124 Å), whereas the *p*-OH group established electrostatic interactions with the C6-carbonyl group of the corresponding *r*G13 and with the C4-primary amine/imine of the second of the *r*C11 and *r*C12. The aliphatic path was oriented toward the *r*C11 to establish induced dipole interactions with the corresponding C4-primary amine/imine portion. Furthermore, the chlorogenic acid’s (−)-quinic acid portion faced the sense strand where the C1-OH made the HB with the *r*T43′s C4-carbonyl portion (*d*_HB_ = 2.803 Å), whereas the C3-OH moiety electrostatically interfered with *r*T43′s C4-carbonyl portion and *r*A44′s C5-NH_2_ portion. The (−)-quinic acid adopted a binding conformation (Figure 3D) in which the *p*-OH group established HB with the C6-carbonyl portion (*d*_HB_ = 2.932 Å) of *r*G41, whereas one of the two *meta* hydroxyl groups (either C3-OH or C5-OH) established HB with the *r*G41′s position N7 (*d*_HB_ = 3.221 Å). Finally, the C1-OH formed HB (*d*_HB_ = 2.546 Å) with a C4-carbonyl moiety of *r*T43 within the sense strand.

#### 2.3.2. Genotoxic and Antigenotoxic Activities of *N. cataria* Leaf Methanol Extract (LME)

LME, administered at 200 mg/kg btw concentration exerted no genotoxic features related to CCl_4_ (the hazardous agent caused 5.52-fold higher tail moment value, 2.56-increment in the % *r*DNA in tail, and 7.78-fold increase in *r*tail length; Table 6: group II vs. group IV), and likewise increased the values of the corresponding markers compared to the control group (Table 6: group IV vs. group I 2.08-fold, 1.37-fold, and 1.22-fold higher values, yet with no significant differences (*p* > 0.05)), compared to *N. cataria* FME, its administration still caused the 1.82-fold increase in the *r*tail moment, as well as 1.13-fold and 1.09-fold increases in % of *r*DNA in tail and tail length, respectively (Table 6, group IV vs. III). 

Nevertheless, the administration of *N. cataria* LME at 100 mg/kg bwt concentration (Table 7: group X) efficiently prevented the CCl_4_-induced *r*DNA damage (causing 1.95-fold lower total score than the hazardous agent; Table 7: group II vs. group X; but 2.08-fold higher one than within the control group; Table 7: group X vs. group I), yet with the lower value of percentage of reduction of 64.80% and still allowing for the slightly higher frequency of comet class *2* related to FME (Table 7: group X vs. group III).

Similarly, rosmarinic acid could in part explain the LME activity, given that it likewise allowed no comets from class *2* to class *4*. Rosmarinic acid at 100 mg/kg bwt concentration protected the *r*DNA from the CCl_4_-caused damage, as proved by the 2.09-fold lower total score than the hazardous agent, but 2.12-fold higher one than within the control group, as well as with the percentage reduction of 67.25% (Table 8). Rosmarinic acid adopted a rather unusual sandwich-like binding conformation in which the aromatic moieties of caffeic acid and dihydrophenillactic acid portions were orthogonally related to each other. In that sense, the caffeic acid *p*ara hydroxyl group established HB with the C4-amine/imine portion of the *r*C12 (*d*_HB_ = 3.308 Å), whereas the *m*-OH was strongly H-bonded with the C6-carbonyl portion (*d*_HB_ = 2.247 Å) of the *r*G41. The aliphatic portion of rosmarinic acid spanned over the *r*DNA double strain in the above-described sandwich-like arrangement, where the free carboxylic group partially faced the *r*G41′s C6-NH_2_ group. Finally, the dihydrophenillactic acid *m*eta hydroxyl group electrostatically attracted the C6-carboxyl portion and position N7 of the *r*G42, whereas the *p*-OH portion similarly attracted the C6-carboxyl portion and N7-nitrogen of the *r*G41.

#### 2.3.3. Genotoxic and Antigenotoxic Activities of *N. cataria* Stem Methanol Extract (SME) 

The extent of liver *r*DNA damage was significantly lower (*p* > 0.05) in animals treated with the *N. cataria* SME at 200 mg/kg bwt concentration compared to those treated with the hazardous agent (which caused 3.50-fold higher tail moment value, 1.42-increment in the % *r*DNA in the tail, and 4.40-fold increase in *r*tail length; Table 6: group I vs. group V). Furthermore, the extract could be harmful to the liver *r*DNA when compared with the reference group (3.28-fold higher *r*tail moment value, 2.47-increment in the % *r*DNA in tail, and 2.15-fold increase in tail length; Table 6, group V vs. group I).

Regarding antigenotoxicity, the lower concentration of 50 mg/kg bwt, compared to CCl_4_, showed the highest %R value of 77.05 (2.39-fold lower total score than the hazardous agent; Table 7: group II vs. group XII, but 1.71-fold higher one than within the control group; Table 7: group XII vs. group I) candidating SME for future potential applications. Regarding the %R value, SME at the lowest concentration (Table 7, group XII) performed similarly to FME at the highest concentration (Table 7, group VIII). Unfortunately, the SME showed a high level of *r*DNA damage (Table 7: group XII, comet class *3*). In conclusion, chlorogenic acid, as the most abundant SME, contained a compound that represents the one most related to the extract activity.

## 3. Materials and Methods

### 3.1. Chemicals and Reagents

LC gradient-grade methanol and formic acid were purchased from J. T. Baker (Deventer, The Netherlands). Methanol, carbon tetrachloride, and dimethyl sulfoxide were obtained from Sigma-Aldrich (Steinheim, Germany). Reference standards of the phenolic compounds were obtained from Sigma-Aldrich (Steinheim, Germany) and Extrasynthese (Genay, France). Hanks balanced salt solution (HBSS), low melting point agarose, normal melting point agarose, bovine serum albumin, phosphate-buffered saline (PBS), lysis buffer, sodium hydroxide (NaOH), ethylenediaminetetraacetic acid (EDTA), absolute alcohol, ethidium bromide and 1,1,3,3-tetraethoxypropane were purchased from Sigma-Aldrich (Steinheim, Germany). 

### 3.2. Plant Material

*N. cataria* flowers, leaves, and stems were collected at the end of the flowering season (June 2014) from natural populations in Serbia, Miljevici village, (altitude 920 m, 43°22′07′′ N, 19°35′25′′ E). The voucher sample was identified, classified, and deposited at the Herbarium of the Department of Biology and Ecology, Faculty of Science, the University of Kragujevac (no. MB03/14).

### 3.3. Preparation of the Extracts

The air-dried (in darkness, at room temperature) aerial parts of flowers, leaves, and stems of *N. cataria* were separately extracted with methanol, using Soxhlet apparatus, at 60 °C for 12 h. Filtered extracts were evaporated to dryness under vacuum at 40 °C using a rotary evaporator and stored in darkness at 4 °C until used.

### 3.4. Liquid Chromatography-ESI-MS/MS Tandem Mass Spectrometry Analysis

Quantitative analysis was performed on Agilent Technologies 1200 HPLC coupled to Agilent Technologies Triple Quadrupole 6420 tandem mass spectrometer (Santa Clara, CA, USA) with ESI source operating in negative ionization mode. MassHunter ver. B.03.01. software (Agilent Technologies, Santa Clara, CA, USA) was used for control of instruments, data acquisition, and data processing. The extracts of flowers, leaves, and stems of *N. cataria* were prepared as methanol solutions (concentrations 0.2 mg/mL). To construct calibration curves, methanolic standard solutions of eighteen phenolic compounds were prepared in the concentration range from 0.00153 to 50.00 μg/mL. Chromatographic separation as well as operating conditions of the mass spectrometer were the same as previously described [61,62]. Briefly, injected volumes of 5 μL of all solutions (analyzed samples or mixtures of standard compounds) were separated on a Zorbax Eclipse XDB-C18 RR 4.6 mm × 50 mm × 1.8 μm (Agilent Technologies, Santa Clara, CA, USA) column (temperature 45 °C, flow rate 1 mL/min). The mobile phase was aqueous formic acid (99.95–0.05%) (A) and methanol (B). Elution program: 0–6 min (30–70% B), 6–9 min (70–100% B), and 9–11 min (100% B). 

Operating conditions of ESI source were: drying gas (N_2_) temperature 350 °C with a flow rate 9 L/min, nebulizer gas (N_2_) 40 psi, and capillary voltage 4 kV. Acquisition of data by previously described parameters (specific fragmentor voltages-V_f_; collision cell voltages-V_c_, selected precursor, and product ions) was performed in dynamic multiple reaction monitoring mode [61,62]. After the construction of calibration curves (peak areas vs. different concentrations of standard compounds), concentrations of phenolic compounds were derived from equations for linear regression. Results were given as means ± SD of three measurements.

### 3.5. N. cataria Secondary Metabolites Retrieval

All the obtained *N. cataria* secondary metabolites were purchased from the PlantMetaChem database (http://www.plantmetachem.com; accessed on 15 March 2020) and used without purification.

### 3.6. Animals and Study Design

Seventy male albino Wistar rats weighing 220 ± 20 g used in this study were obtained from the Animal House of Military Medical Academy, Belgrade, Serbia and acclimatized for three days before the experiment. Maintenance was under a 12 h light–dark cycle, with food and water available *ad libitum*. All animal procedures were approved by the Ethical Committee of the Faculty of Science, the University of Kragujevac, Number of Ethical Approval 2-06/2022, which acts following the relevant Serbian guidelines, including the Guidelines for the Care and Use of Laboratory Animals and Law on Animal Welfare (“Official Gazette of Republic of Serbia”, no. 810 41/09) and the European Directive for the Welfare of Laboratory Animals Directive 2010/63/EU.

Male albino rats were equally divided into fourteen groups consisting of five animals in each group and treated orally for five days, as follows: group I (normal control) was daily given distilled water (500 mL per animal) and then intraperitoneally (*i.p*.) injected with 1 mL/kg body weight (bwt) of commercial olive oil (Monini Olio Extra Vergine di Oliva). Group II, positive control, was orally given distilled water for five days, and then *i.p*. injected with a single dose of 1 mL/kg body weight CCl_4_ (1:1 mixture in olive oil) [35]. For the hepatotoxicity and genotoxicity studies, the animals in groups III-V separately received orally a single dose of 200 mg/kg bwt of FME, LME, and SME, each dissolved in commercial olive oil, for five days. For the hepatoprotective and antigenotoxic studies, the animals in groups VI-VIII were administrated with FME of *N. cataria* at 50, 100, and 200 mg/kg bwt, dissolved in commercial olive oil; animals in groups IX-XI were treated with LME of *N. cataria* at 50, 100 and 200 mg/kg bwt, dissolved in commercial olive oil, respectively, whereas the animals in in groups XII-XIV were treated with LME of *N. cataria* at 50, 100 and 200 mg/kg bwt, dissolved in commercial olive oil. On the last day of the treatment, the animals of groups II-XI received *i.p*. a single dose of 1 mL/kg body weight CCl_4_ (1:1 mixture in olive oil). Twenty-four hours after CCl_4_ injection, all the animals were anesthetized with ethyl ether and afterward sacrificed, and their livers and blood (from the abdominal vein) were collected immediately in non-heparinized tubes.

The additional experiments were conducted using the quantified metabolites of *N. cataria* using a similar experimental setup as for hepatoprotective and antigenotoxic studies: each new group of the experimental animals (five animals per group) was treated per os with a single dose of the investigated compound, either 50, 100, and 200 mg/kg body weight, before the *i.p.* administration of CCl_4_ (1 mL/kg bwt).

### 3.7. Determination of Hepatoprotective Activity

#### 3.7.1. Measurement of Serum Toxicity Markers

The serum for determination of biochemical parameters, aspartate transaminase (AST), alanine transaminase (ALT), alkaline phosphatase (ALP), and γ-glutamyltransferase (γ-GT) was prepared by the Quick method [63], immediately immersed in liquid nitrogen, and stored at −80 °C until use. The catalytic activities of AST and ALT at 340 nm, as well as that of ALP and γ-GT at 405 nm, were determined by UV–VIS kinetic methods according to recommendations of the Expert Panel of the IFCC (International Federation of Clinical Chemistry) [64,65,66,67]. Total protein concentrations were determined according to the Lowry method using bovine serum albumin as the standard [68]. All kinetic and colorimetric measurements were performed using a Dynamica HALO DB-20 UV-VIS spectrophotometer.

#### 3.7.2. Measurement of Liver Homogenate Antioxidant Markers

Rat liver samples were homogenized in phosphate buffer (5 mM, pH 7.4) to yield a 10% (*w/v*) homogenate and then centrifuged at 4000 rpm for 15 min at 4 °C. The supernatants were used to estimate the catalytic activity of superoxide dismutase (SOD) [39], the level of TBARS [40], the catalytic activity of catalase (CAT) [41], and the concentration of reduced glutathione (GSH) [41] by the colorimetric method. Total protein concentrations were determined according to the Lowry method [68]. All colorimetric measurements were performed using a Dynamica HALO DB-20 UV-VIS spectrophotometer.

### 3.8. Determination of Genotoxic Activity

#### 3.8.1. Detection of DNA Damage

The alkaline single cell gel electrophoresis assay was performed as described in the literature [58]. The liver samples were excised, and small fragments of livers were transferred on ice. The fragments were washed, minced and suspended into 1 mL ice-cold Hank’s balanced salt solution (HBSS; 0.14 g/L CaCl_2_, 0.4 g/L KCl, 0.06 g/L KH_2_PO_4_, 0.1 g/L MgCl_2_ × 6H_2_O, 0.1 g/L MgSO_4_ × 7H_2_O, 8.0 g/L NaCl, 0.35 g/L NaHCO_3_, 0.09 g/L Na_2_HPO_4_ × 7H_2_O, 1.0 g/L D-glucose, 20 mM EDTA and 10% DMSO). A fresh mincing solution was added, and the liver samples were minced again into finer pieces. From the liver cell suspension, 10 μL was mixed with 75 μL of 1% low melting point agarose (in Ca^+2^ and Mg^+2^ free PBS, pH 7.4). From the final cell–agarose suspension, 85 μL was spread over the microscope slide pre-coated with 1.5% normal melting point agarose in PBS buffer and covered with cover-glass. The gel was allowed to set at 4°C, and cells were lysed for a period of at least 2 h in cold lysis buffer (2.5 M NaCl, 100 mM EDTA, 10 mM Tris, 1% Triton X-100, 10% DMSO, pH 10.0). After lysis, the slides were then alkaline-unwound, and then, electrophoresis was carried out using the electrophoresis buffer (300 mM NaOH and 1 mM EDTA, pH > 13) at 4 °C for 30 min at 25 V with the current adjusted to 300 mA. After electrophoresis, the slides were then neutralized with neutralization buffer (0.4 M Tris-HCL, pH 7.4), three times for 5 min and fixed for 5 min in absolute alcohol, air-dried, and stored at room temperature. Immediately before analysis, the slides were stained with 75 μL ethidium bromide (20 μg/mL) and covered with a cover slip.

#### 3.8.2. Data Scoring and Photomicrographs

Comets were visualized and captured with 400× objective lens of fluorescence microscope Nikon (Ti-Eclipse) attached to CCD camera. One hundred comet images per slide were randomly captured and analyzed. Comets without heads and with almost all the DNA in the tail or with a wide tail were excluded from the analysis since they could represent dead cells. Cells that did not overlap and had a clear margin surrounding them were scored [69].

Three parameters were selected as indicators of DNA damage: tail moment, tail length, and % DNA in comet tail [70]. According to Collins, comets were classified into five categories defined as types *0*, *1*, *2*, *3* and *4* (no or low damage, low, medium and long DNA migration, and the highest level of degradation, respectively) [70]. The total comet score and the percentage reduction (%R) in the total comet score was calculated using Formula (1):(1)$$\%R=\frac{\mathrm{mean}\;\mathrm{total}\;\mathrm{score}\;\mathrm{in}\;A\;-\;\mathrm{mean}\;\mathrm{total}\;\mathrm{score}\;\mathrm{in}\;B}{\mathrm{mean}\;\mathrm{total}\;\mathrm{score}\;\mathrm{in}\;A\;-\;\mathrm{mean}\;\mathrm{total}\;\mathrm{score}\;\mathrm{in}\;C}\;\times \;100$$
where *A* is the mean of total score in positive control, *B* mean of total score in pretreatment with the extracts prior to CCl_4_, and *C* is the mean of total score in negative control [71,72]. 

### 3.9. Statistical Analysis

Data were analyzed using SPSS statistical software package (version 13.0). One-way analysis of variance (ANOVA) followed by T3 Dunnett test or with Bonferroni test for post hoc comparison between controls and treated groups was used. The results were considered to be statistically significant at *p* < 0.05.

### 3.10. Structure-Based Studies

#### 3.10.1. Ligands Modeling 

The naturally occurring compound structures were collected from the PlantMetaChem database. Each of the compounds was retrieved in Structure Data File (SDF) format, added of hydrogens and charges (the Gasteiger model) at the physiological pH, by means of the OpenBabel toolkit [73] and energy minimized by virtue of OpenBabel’s obconformer module: the lowest energy conformer was selected from 100 conformations after 100 geometry optimization steps using a Monte Carlo search. Optimized compounds were stored in mol2 format for further manipulation.

MDA was drawn and modeled using the Chemaxon’s msketsh module [74] by means of molecular mechanics optimization upon which the correct protonation at pH 7.4 was assigned. 

#### 3.10.2. *N. cataria* Secondary Metabolites Bioactive Conformations Determination within the rCATCmpd I

The *R. norvegicus* catalase compound I was prepared from the three-dimensional structure of rat erythrocyte catalase retrieved from AlphaFold database (ID: P04762 (CATA_RAT)) (https://alphafold.ebi.ac.uk; accessed on 15 March 2020) [75,76]. The structure was loaded into UCSF Chimera [76] and visually inspected. The protein was further improved by adding hydrogen atoms using the embedded leap module of Amber 12 suite [77] upon which the correct hydrogen atoms, appropriate for pH 7.4, were assigned to each amino acid residue. Protein was then energy minimized as follows: through the leap module, they were solvated with water molecules (TIP3P model, SOLVATEOCT Chimera command) in a box extending 10 Å in all directions, neutralized with either Na^+^ or Cl^−^ ions, and refined by a single point minimization using the Sander module of Amber suite with maximum 1000 steps of the steepest-descent energy minimization and maximum 4000 steps of conjugate-gradient energy minimization, with a non-bonded cutoff of 5 Å. From the properly prepared complex, the human catalase compound I was created by means of the Chimera Tools, invoking the Build Structure module and Modify Structure functions, by changing the appropriate hybridization of iron and oxygen within the oxoferryl porphyrin π-cation radical state. The newly formed structure was then energy-minimized following the exact procedure as described above, and afterward used for molecular docking.

The modeled enzyme was further manipulated by removing the nonpolar hydrogen atoms, while Gasteiger charges and solvent parameters were added. Afterward, the obtained *N. cataria* secondary metabolites were docked using the following experimental setup. The rigid root and rotatable bonds were defined using AutoDockTools [77]. The docking was performed with AutoDock 4.2 [77]. Before actual docking, the grid spacing had been determined by means of the modified hem ring within the rat catalase compound I: the XYZ coordinates (in Ångströms) of the cuboid grid box used for the computation were X_min_/X_max_ = 3.329/29.969, Y_min_/Y_max_ = 28.466/52.199, Z_min_/Z_max_ = 44.984/67.222 to embrace the space spanning 10 Å in all three dimensions. The Lamarckian Genetic Algorithm was used to generate the bioactive conformations within the active site. The global optimization started with a population of 200 randomly positioned individuals, a maximum of 1.0 × 10^6^ energy evaluations, and a maximum of 27,000 generations. In total, 100 runs were performed with RMS Cluster Tolerance of 0.5 Å.

#### 3.10.3*. N. cataria* Secondary Metabolites Bioactive Conformations Determination within the rTopIIα

Considerations of genotoxic and antigenotoxic mechanistic studies require a profound knowledge of *R. norvegicus* TopIIα structure. As mentioned before, *r*TopIIα is a homodimer consisting of three independent but merged-in-action subregions called *rN*-terminal, *r*DNA-binding and cleavage, and *rC*-terminal domains [60]. The structure of each *rN*-terminal bears the *r*ATPase domain with the function to accept and hydrolyze an *r*ATP molecule prior and upon the *r*DNA ligation and religation, respectively. The interaction of the *r*ATPase with *r*ATP induces domain dimerization or separation, resulting in the closing and opening of the *rN*-gate, respectively [60]. The *r*DNA-binding and cleavage region is formed by the *r*TOPRIM (TOpoisomerase/PRImase) domain, which contains *r*Mg^2+^ involved in the *r*DNA cleavage mechanism, and *r*WHD (winged-helix domain) area, bearing the active site tyrosine. In the end, the *rC*-terminus itself acts as *C-*gate [60]. 

Therefore, the structure-based studies within *r*TopIIα’s DNA-binding and cleavage region were performed on a homology-modeled enzyme, using the identical experimental setup as described elsewhere [60] by means of AutoDock Vina [78]. The enzyme was prepared upon and was loaded into UCSF Chimera software [79] and improved by adding hydrogen atoms using the leap module of Amber 12 suite [80], upon which the correct protonation at pH = 7.4 was assigned to each amino acid and nucleic acid residue by means of the Antechamber module of Amber 12 suite. The complex was then solvated (SOLVATEOCT command) in a box entering 10 Å with water molecules (TIP3 model), neutralized with Na^+^ and Cl^−^ ions, and refined by a single point minimization using the Sander module of AMBER suite in 1000 steps of the steepest-descent energy minimization followed by 4000 steps of conjugate-gradient energy minimization, with a non-bonded cutoff of 5 Å. 

For Vina, the XYZ coordinates (in Ångströms) of the cuboid grid box used for the computation were X_min_/X_max_ = 4.543/26.767, Y_min_/Y_max_ = 31.324/57.443, Z_min_/Z_max_ = 41.324/75.656 to embrace the space spanning 10 Å in all three dimensions. The docking simulations were carried out with an energy range of 10 kcal/mol and exhaustiveness of 100 with RMS Cluster Tolerance of 0.5 Å. The output comprised 20 different conformations for every receptor considered. 

## 4. Conclusions

The enclosed report presents for the first time the LC-ESI-MS/MS analysis of phenolic compounds, as well as hepatoprotective and antigenotoxic activities of the methanol extracts of various *N. cataria* plant organs. The FME was characterized by high concentrations of quercitrin, chlorogenic acids, and quinic acid, while the most abundant compounds in the LME and SME were rosmarinic and chlorogenic acid, respectively. 

Extracts were assayed in vivo with Wistar rats to investigate liver intoxication with CCl_4_, which through metabolic activation has generated hepatotoxic metabolites, namely trichloromethyl (*r*CCl_3_^•^) and trichloromethylperoxy (*r*CCl_3_OO^•^) radicals. While entering the hepatocytes, the radicals induced the malondialdehyde (*r*MDA)-based peroxidation of polyunsaturated fatty acids in the membrane of the hepatocyte. The examined extracts protected the membrane of the hepatocyte from disruption, as evaluated by the status of liver antioxidant (*r*SOD, *r*TBARS, *r*CAT, and *r*GSH) and toxicity markers (*r*AST, *r*ALT, *r*ALP, and *r*γ-GT), where the *N. cataria* FME, LME, and SME at 200, 100, and 50 mg/kg bwt concentrations, respectively, could be considered as potential candidates as supplements for hepatoprotection. Special attention was also given to the secondary metabolites present in the extracts that were investigated similarly to the extracts. All the tested pure compounds exerted a profile related to that of extracts leading to assess the most abundant compounds as bearers of extract properties, although with some synergistics still to be investigated. A computational procedure supported the experimental data and indicated that all secondary metabolites can decrease the catalytic activity of *R. norvegicus r*CAT, based on SB investigation in terms of their interaction with the enzyme’s a porphyrin–radical cation, called *R. norvegicus* catalase compound I (*r*CATCmpd I), resulting in the appearance of *r*CATCmpd II, at the same time transforming themselves to the corresponding semiquinone radicals forcing the formation of new amounts of *r*H_2_O_2_. 

At the *r*DNA level, the *r*CCl_3_˙ and *r*CCl_3_OO˙ radicals may participate in the generation of *r*MDA that could interfere with the liver *r*DNA and form *r*dG, *r*dA, and *r*dC adducts, subsequently oxidized to *r*M_1_G, *r*M_1_A, and *r*M_1_C with a fate to endue as *rN*^2^-oxopropenyl-dG, *rN*^2^-oxopropenyl-dA, and *rN*^2^-oxopropenyl-dC adducts causing the *R. norvegicus* TopIIα-catalyzed *r*DNA-DNA inter-strand cross-links or *r*DNA-protein inter-strand crosslinks. In that sense, the results of antigenotoxic activity by means of comet assay provided evidence that all tested extracts cause a decrease in the level of *r*DNA damage induced by the CCl_4_, lowering the total score and percentage of reduction of *r*DNA and forbidding the appearance of comet classes associated with the severe damage. Within the extracts, the found secondary metabolites exerted antigenotoxicity upon occupying the *r*DNA binding and cleavage domain of *R. norvegicus* TopIIα, for which they denied the *r*MDA to form the aberrant adducts. 

With the fact that *N. cataria* is extensively used in pet toys, together with its significant role in traditional medicine, the above-mentioned results for the first time confirm the plant’s safety from the point of absence of hepatotoxicity and genotoxicity and its possible use as a hepatoprotective and antigenotoxic agent.

## Data Availability

Not applicable.

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
