# Peer review of "Chemical Composition of Various Nepeta cataria Plant Organs’ Methanol Extracts Associated with In Vivo Hepatoprotective and Antigenotoxic Features as well as Molecular Modeling Investigations"

_plants, 2022, doi:10.3390/plants11162114_

Round 1

Reviewer 1 Report

The manuscript "Chemical composition of various Nepeta cataria plant organs’ methanol extracts associated with the in vivo hepatoprotective and antigenotoxic features, and molecular modeling investigations" is an interesting and well written paper. I have only one recommendation consisting to add a conclusion section after the results and discussion section.

Author Response

Reviewer 1

The manuscript "Chemical composition of various Nepeta cataria plant organs’ methanol extracts associated with the in vivo hepatoprotective and antigenotoxic features, and molecular modeling investigations" is an interesting and well written paper. I have only one recommendation consisting to add a conclusion section after the results and discussion section.

Dear Sir/Madam,

Thank you very much for reviewing the enclosed manuscript. All of your comments and remarks were carefully considered and below you can find the list of responses to each of the raised questions. All of your suggestions significantly improved the quality of the manuscript.

Thank you very much for this valuable suggestion. We agree with the reviewer that the Conclusion section should be more meaningful after the Results and Discussion, but according to Authors guidelines for preparing manuscripts ready to submit in Plants journal, Conclusion section is required after the Materials and Methods.

Reviewer 2 Report

Dear Author,

The manuscript has potential but before being able to have a decision I would like to address some questions / requests.

Frist of all, for me is not clear what product received the animals. In many parts of the manuscript is written the methanolic extract, then in the methods they were concentrated to vacuum (up to what point? Powder?). than they were dissolved in olive oil… Please describe the final product. 

The term fibroses is often used, but no histopathological examination was done (I do not understand why it is missing, in my opinion the lack of morphological prove is major drawback of the study). Please add some data to prove the fibrosis or replace it with other term, as hepatotoxicity or similar.

In the future, I suggest some pilot toxicity studies to prove that the extract have no toxicity following standardized procedures like OECD protocols (not only at the liver tissue). I would also recommend avoiding using so many independent variables in the same study (and so many experimental groups), which makes the study difficult to conduct. In my opinion testing three doses of each extract uses an unnecessary number of animals. One extract with better phytochemical composition would be enough for the three-dose testing for dose-dependent effect.

I suggest presenting more clearly the actual results and not mixing them in a confusing manner with the bibliographical findings. I have given some examples below.

I added some punctual remarks I hope you will find them helpful.

Row 64 traditional medicine of what country / region.

Row 100- to row 109 I don’t understand what this text is… probably some instructions left…

Row 295 for me the table 3 it is not clear. These animals are additional to those receiving the extracts? There is nothing in the methods about animals receiving quinic acid/ protocatechinic acid etc. The same comment for table 5.

Row 808 It Is mandatory to mention the number of the approval. Please notice that Directive 86/609/EEC is no longer in force since 2010. Additionally, the EU legislation is relevant for EU member states and Norway. I would recommend to indicate the national regulations only. How were those doses established? Any preliminary toxicity tests? Bibliography? 

Row 374 the hepatocytolysis suggested by elevated AST and ALP should not be confused with cirrhosis. The only way to prove cirrhosis is histopathology. Please correct all the manuscript accordingly.

row 368 the affirmation “CCl4-induced formation of rCCl3•, rCCl3OO•, and rO2•- radicals” is not supported by the present findings as those reactive species were not measured. As a general suggestion, I recommend not mixing the actual findings to discussion in the same phrase. The bibliographical information should be always supported by bibliographical citation.

812 – I did no find what FME, LME and SME mean. Please include the full name when mention the first time and in the Tables legend.  

Row 816 how much distilled water did receive the animals in the control group? Did you inject olive oil intraperitoneally? How much? Is it a reagent made from the company or from the marked?  

Row 826 Please describe the method used for the blood collection, and how were the animals euthanized.

Author Response

Reviewer 2

Dear Author,

The manuscript has potential but before being able to have a decision I would like to address some questions / requests.

Dear Sir/Madam,

Thank you very much for reviewing the enclosed manuscript. All of your comments and remarks were carefully considered and below you can find the list of responses to each of the raised questions. All of your suggestions significantly improved the quality of the manuscript.

Frist of all, for me is not clear what product received the animals. In many parts of the manuscript is written the methanolic extract, then in the methods they were concentrated to vacuum (up to what point? Powder?). than they were dissolved in olive oil… Please describe the final product.

Thank you very much for this valuable questions. Filtered extracts were evaporated to dryness under vacuum at 40 °C using a rotary evaporator, and stored in darkness at 4 °C until used.

Regarding administration to experimental animals, the dry extracts were disolved in commercially available olive oil and were given to the animals.- See section 3.6. ``Animals and study design``.

The term fibroses is often used, but no histopathological examination was done (I do not understand why it is missing, in my opinion the lack of morphological prove is major drawback of the study). Please add some data to prove the fibrosis or replace it with other term, as hepatotoxicity or similar.

Thank you very much for this valuable comment. We agree with the reviewer that histopathological examination would improve the quality of the results and overall presentation. However, the discussion is now adopted in terms of reviewer’s comments by means of using hepatotoxicity instead of fibrosis.

In the future, I suggest some pilot toxicity studies to prove that the extract have no toxicity following standardized procedures like OECD protocols (not only at the liver tissue).

Thank you very much for this valuable comment and suggestion for a future work. We agree with the reviewer that OECD protocols would improve the quality of our research and overall presentation of the results. Of course, that in future we will  conduct OECD protocols.

I would also recommend avoiding using so many independent variables in the same study (and so many experimental groups), which makes the study difficult to conduct. In my opinion testing three doses of each extract uses an unnecessary number of animals. One extract with better phytochemical composition would be enough for the three-dose testing for dose-dependent effect.

Thank you very much for this valuable comment and suggestion for a future work. We agree with the reviewer that using many independent variables and experimental animals is sufficient. However, testing of a given extract in several different doses is somehow a standard protocol in papers with similar concept:

Fatima H, Shahid M, Jamil A, Naveed M, Therapeutic Potential of Selected Medicinal Plants Against Carrageenan Induced Inflammation in Rats, Dose-Response, 2021, 19 (4), 15593258211058028. DOI: 10.1177/15593258211058028

Pal RS, Mishra A, Evaluation of Acute Toxicity of the Methanolic Extract of Dhatryadi Ghrita in Wistar Rats. The Open Pharmacology Journal, 2019, 9, 1-4.

Al Bashera, M.; Mosaddik, A.; El-Saber Batiha, G.; Alqarni, M.; Islam, M.A.; Zouganelis, G.D.; Alexiou, A.; Zahan, R. In Vivo and In Vitro Evaluation of Preventive Activity of Inflammation and Free Radical Scavenging Potential of Plant Extracts from Oldenlandia corymbosa L. Appl. Sci. 2021, 11(19), 9073; https://doi.org/10.3390/app11199073

Matić S, Stanić S, Bogojević D, Vidaković M, Grdović N, Arambašić J, Dinić S, Uskoković A, Poznanović G, Solujić S, Mladenović M, Marković J, Mihailović M. Extract of the plant Cotinus coggygria Scop. attenuates pyrogallol-induced hepatic oxidative stress in Wistar rats. Canadian Journal of Physiology and Pharmacology, 2011, 89 (6), 401-411. DOI: 10.1139/y11-043; ISSN: 0008-4212

Katanić J, Matić S, Pferschy-Wenzig EM, Kretschmer N, Boroja T, Mihailović V, Stanković V, Stanković N, Mladenović M, Stanić S, Mihailović M, Bauer R. Filipendula ulmaria extracts attenuate cisplatin-induced liver and kidney oxidative stress in rats: In vivo investigation and LC-MS analysis. Food and Chemical Toxicology, 2017, 99, 86-102. DOI: 10.1016/j.fct.2016.11.018; ISSN: 0278-6915

Mihailović V, Mihailović M, Uskoković A, Arambašić J, Mišić D, Stanković V, Katanić J, Mladenović M, Solujić S, Matić S. Hepatoprotective effects of Gentiana asclepiadea L. extracts against carbon tetrachloride induced liver injury in rats. Food and Chemical Toxicology, 2013, 52, 83-90. DOI: 10.1016/j.fct.2012.10.034; ISSN: 0278-6915

Therefore, deferent doses were here used as a sort of benchmark for extracts’ safety. Still, with the awareness that analysing of three extracts in three different doses would be hard task to properly do and would be very hard to follow by a reader, the focus was given only on the most promising (best performing) concentrations of given extracts and comparison with others of the comparable pharmacological profile. Therefore, for each extract only the best concentration in pharmacological terms was discussed. In our opinion, it was the optimal way of presenting the results of this study, proving the reader ad hoc the most important results. Formulating one extract with an optimal chemical composition would be indeed beneficial and will be likely done in the continuation of this study.

I suggest presenting more clearly the actual results and not mixing them in a confusing manner with the bibliographical findings. I have given some examples below.

Thank you very much for this valuable comment and suggestion for a future work. However, with all due respect to the reviewer’s opinion, we cannot agree that the herein results are mixed in a confusing manner with the bibliographical findings. The introductory part clearly summarises the known pharmacology of N. cataria. The herein reported results concerning the hepatoprotective/hepatotoxic and genotoxic/antigenotoxic features of N. cataria, alongside with the inner mechanisms of action, have been to the best of authors knowledge unreported so far, and therefore present a new contribution, which cannot be mixed and confused with the previous bibliographical findings.

I added some punctual remarks I hope you will find them helpful.

Row 64 traditional medicine of what country / region.

Thank you very much for this valuable comment. The countries and regions in whin N. Cataria is used as traditional remedy, namely France, England and region of North America, are now included in the main text. This statement was supported by [15].

Sharma, A.; Nayik, G.A.; Cannoo, D.S. Pharmacology and Toxicology of Nepeta cataria (Catmint) Species of Genus Nepeta: A Review. In: Ozturk M., Hakeem K. (eds) Plant and Human Health, Volume 3. Springer, Cham. https://doi.org/10.1007/978-3-030-04408-4_13.

Row 100- to row 109 I don’t understand what this text is… probably some instructions left…

Thank you very much for this valuable observation. The listed lines were indeed the instructions left from the manuscript preparation and are now removed.

Row 295 for me the table 3 it is not clear. These animals are additional to those receiving the extracts? There is nothing in the methods about animals receiving quinic acid/ protocatechinic acid etc. The same comment for table 5.

Thank you very much for these valuable questions. You are correct, pure compounds were administered to new experimentals animals, and that compounds were subjected to the same experimental protocols as extracts by means of concentrations and assay kits, which is described in the last paragraph of section 3.6:

The additional experiments were conducted using the quantified metabolites of N. cataria using a similar experimental setup as for hepatoprotective and antigenotoxic studies: each new group of the experimental animals (five animals per group) was treated per os with a single dose of the investigated compound, either 50, 100, and 200 mg/kg body weight, before the i.p. administration of CCl4 (1 mL/kg bwt).

Row 808 It Is mandatory to mention the number of the approval. Please notice that Directive 86/609/EEC is no longer in force since 2010. Additionally, the EU legislation is relevant for EU member states and Norway. I would recommend to indicate the national regulations only.

Thank you very much for this valuable observations. At the present moment, as well during the period when the experimental part of this manuscript was performed, the number of approval was (is) unavailable. However, all animal procedures were approved by the Ethical Committee of the Faculty of Science, the University of Kragujevac, which acts following the relevant Serbian guidelines, including the Guidelines for the Care and Use of Laboratory Animals and Law on Animal Welfare ("Official Gazette of Republic of Serbia", No. 810 41/09) and the European Directive for the Welfare of Laboratory Animals Directive 2010/63/EU.

This is a standard regulative used at University of Kragujevac, also used in other representative publications:

Natalija Arsenijevic, Dragica Selakovic, Jelena S. Katanic Stankovic, Vladimir Mihailovic, Slobodanka Mitrovic, Jovana Milenkovic, Pavle Milanovic, Miroslav Vasovic, Aleksandra Nikezic, Olivera Milosevic-Djordjevic, Marko Zivanovic, Nenad Filipovic, Vladimir Jakovljevic, Nemanja Jovicic, Gvozden Rosic, Variable neuroprotective role of Filipendula ulmaria extract in rat hippocampus, Integr. Neurosci. 2021, 20(4), 871–883; https://doi.org/10.31083/j.jin2004089

Rade Vukovic, Dragica Selakovic, Jelena S. Katanic Stankovic, Igor Kumburovic, Nemanja Jovicic, Gvozden Rosic, Alteration of Oxidative stress and apoptotic markers alterations in the rat prefrontal cortex influence behavioral response induced by cisplatin and N-acetylcysteine in the tail suspension test, J. Integr. Neurosci. 2021, 20(3), 711–718; https://doi.org/10.31083/j.jin2003076

Kurtanović N, Tomašević N, Matić S, Mitrović MM, Kostić DA, Sabatino M, Antonini L, Ragno R, Mladenović M. Human estrogen receptor α antagonists, part 2: Synthesis driven by rational design, in vitro antiproliferative, and in vivo anticancer evaluation of innovative coumarin-related antiestrogens as breast cancer suppressants. Eur J Med Chem. 2022 Jan 5;227:113869. doi: 10.1016/j.ejmech.2021.113869.

Kurtanović, N.; Tomašević, N.; Matić, S.; Proia, E.; Sabatino, M.; Antonini, L.; Mladenović, M.; Ragno, R. Human Estrogen Receptor Alpha Antagonists, Part 3: 3-D Pharmacophore and 3-D QSAR Guided Brefeldin A Hit-to-Lead Optimization toward New Breast Cancer Suppressants. Molecules 2022, 27, 2823. https://doi.org/10.3390/molecules27092823

How were those doses established? Any preliminary toxicity tests? Bibliography? 

Thank you very much for this valuable questions. The doses were established arbitrarlilly, given that no similar experiments were done before for N. cataria, at least to the best of authors knowlegde. At this stage of investigation, three doses were used to find the benchmark for save and toxic concentrations, which we certanly succeded for each of the examined extract.

Following the fact that no similar experiments were done for N. cataria, no preliminary studues were performed. As a matter of fact, we believe that preliminary studies are not the optimal way to investigate plants extract (or to conduct any study, in general), and that the full-scale well designed study, with optimised assays, is appropriate when the specific pharmacological properties are investigated, as in the herein study.

Row 374 the hepatocytolysis suggested by elevated AST and ALP should not be confused with cirrhosis. The only way to prove cirrhosis is histopathology. Please correct all the manuscript accordingly.

Thank you very much for this valuable observations. We agree with the reviewer and, acordingly, the term hepatocellular cirhossis was changed to hepatotoxicity.

row 368 the affirmation “CCl4-induced formation of rCCl3•, rCCl3OO•, and rO2•- radicals” is not supported by the present findings as those reactive species were not measured. As a general suggestion, I recommend not mixing the actual findings to discussion in the same phrase. The bibliographical information should be always supported by bibliographical citation.

Thank you very much for this valuable observations. We agree with the reviewer and, acordingly, the affirmation “CCl4-induced formation of rCCl3•, rCCl3OO•, and rO2•- radicals” was omitted from the main text.

812 – I did no find what FME, LME and SME mean. Please include the full name when mention the first time and in the Tables legend.  

Thank you very much for this valuable observation. The FME, LME and SME are the abbreviations used for flowers methanol extract, leaves methanol extract, and stems methanol extracts, respectively. These abbreviations were indicated in Abstract

This report summarizes the chemical composition analysis of Nepeta cataria L. flowers, leaves, and stems methanol extracts (FME, LME, SME, respectively) as well as their hepatoprotective and antigenotoxic features in vivo and in silico.

as well as in section 2.1.

In agreement with previously published qualitative composition of N. cataria aerial parts [22], seventeen phenolic compounds and the quinic acid (an intermediate in plant phenolics biosynthesis), were selected as standards (Table 1) for the quantitative investigation of flowers (FME), leaves (LME), and stems (SME) methanol extracts  (Figure 1A, 1B, and 1C, respectively) by means of LC-ESI-MS/MS analysis.

Row 816 how much distilled water did receive the animals in the control group?

Thank you very much for this valuable question. Herein, the water was given to the experimental animals just for fulfiling their physiological needs by consuming it. Each animal was given an amount of 500 mL for free uptake, without monitoring how much water each animal took. Supplying the water was not a part of per os administration of extract, and therefore cannot be considert as a drawback of the experimental protocol.

 Did you inject olive oil intraperitoneally? How much? Is it a reagent made from the company or from the marked? 

Thank you very much for this valuable question. Within the section 3.6. it is now clearly stated how the extracts were administered:

Male albino rats were equally divided into fourteen groups consisting of five animals in each group and treated orally for five days, as follows: group I (normal control) was daily given distilled water (500 mL per animal) and then intraperitoneally (i.p.) injected with 1 mL/kg body weight (bwt) of commercial olive oil (Monini Olio Extra Vergine di Oliva). Group II, positive control, was orally given distilled water for five days, and then i.p. injected with a single dose of 1 mL/kg body weight CCl4 (1:1 mixture in olive oil) [32]. For the hepatotoxicity and genotoxicity studies, the animals in groups III-V separately received orally a single dose of 200 mg/kg bwt of FME, LME, and SME, respectively, each dissolved in commercial olive oil, for five days. For the hepatoprotective and antigenotoxic studies, the animals in groups VI-VIII were administrated with FME of N. cataria at 50, 100, and 200 mg/kg bwt, dissolved in commercial olive oil, respectively, animals in groups IX-XI were treated with LME of N. cataria at 50, 100 and 200 mg/kg bwt, dissolved in commercial olive oil, respectively, whereas the animals in in groups XII-XIV were treated with LME of N. cataria at 50, 100 and 200 mg/kg bwt, dissolved in commercial olive oil, respectively. On the last day of the treatment, the animals of groups II-XI received i.p. a single dose of 1 mL/kg body weight CCl4 (1:1 mixture in olive oil).  Twenty-four hours after CCl4 injection, all the animals were anesthetized with ethyl ether and afterward sacrificed, and their livers and blood (from the abdominal vein) were collected immediately in non-heparinized tubes.

The olive oil used in the experiment was commercial.

Row 826 Please describe the method used for the blood collection, and how were the animals euthanized.

Thank you very much for this valuable question. The detailed explanation is ecnlosed and incorporated in the main text: Twenty-four hours after CCl4 injection, all the animals were anesthetized with ethyl ether, afterward sacrificed, and their livers and blood (from the abdominal vein) were collected immediately in non-heparinized tubes.

Reviewer 3 Report

This revised version of manuscript is well-written and concise.

I think that the authors have a bit complicated their study by studying the organs of the plant separately, for a simple reason: because the plant is used entirely in the folk medicine. However, it is good to know the chemical profile of each organ when the organs could be easily separated "most likely the trees organs but not the herbs". One important class of Catnip was not found in the extract which is the iridoid class which includes the famous nepetalactone, as cited in the reference number 8 in your manuscript.

Mišić, D.; Šiler, B.; Gašić, U.; Avramov, S.; ŽivkoviĆ, S.; Živković, J.N.; Milutinović, M.; Tešić, Ž. Simultaneous 1077 UHPLC/DAD/(+/-)HESI-MS/MS Analysis of phenolic acids and nepetalactones in methanol extracts of Nepeta species: A possi- 1078 ble application in chemotaxonomic studies. Phytochem. Anal. 2015, 26(1), 72-85. Doi.org/10.1002/pca.2538

How can you explain this absence?

Please include in the introduction this important class of compounds "iridoid and iridoid glycosides"  in Catnip which attract the domestic animals specially the cats. I can recommend the following references

Lichman, B. R., Godden, G. T., Hamilton, J. P., Palmer, L., Kamileen, M. O., Zhao, D., ... & O’Connor, S. E. (2020). The evolutionary origins of the cat attractant nepetalactone in catnip. Science advances, 6(20), eaba0721.

Tagawa, M., & Murai, F. (1980). A new iridoid glucoside, nepetolglucosylester from Nepeta cataria. Planta medica, 39(06), 144-147.

Xie, S., Uesato, S., Inouye, H., Fujita, T., Murai, F., Tagawa, M., & Shingu, T. (1988). Absolute structure of nepetaside, a new iridoid glucoside from Nepeta cataria. Phytochemistry, 27(2), 469-472.

Author Response

Reviewer 3

Dear Sir/Madam,

Thank you very much for reviewing the enclosed manuscript. All your comments and remarks were carefully considered and below you can find the list of responses to each of the raised questions. All your suggestions significantly improved the quality of the manuscript.

This revised version of manuscript is well-written and concise.

Thank you very much for your comment.

I think that the authors have a bit complicated their study by studying the organs of the plant separately, for a simple reason: because the plant is used entirely in the folk medicine. However, it is good to know the chemical profile of each organ when the organs could be easily separated "most likely the trees organs but not the herbs".

Thank you very much for your comments. We believe that chemical and biological examinations of herb plant organs were scientifically justified. Of course, folk medicine in some cases uses the whole plant (but in many situations folk medicine recommends the usage of leaves, flowers, roots…). Since biosynthesis and accumulation of bioactive constituents were different in different plant parts, their biological examinations were a reasonable and logical scientific step in defining and selecting the most active plant organ extracts (of course also in a possible introduction in ethnomedicine).

One important class of Catnip was not found in the extract which is the iridoid class which includes the famous nepetalactone, as cited in reference number 8 in your manuscript. How can you explain this absence?

Thank you very much for this question. Our research was focused on the impact of phenolics and flavonoids on hepatoprotective and antigenotoxic activities. Of course, our results clearly demonstrated differences in the activities of different plant organ extracts (due to the characteristic composition of phenolics and flavonoids).

Please include in the introduction this important class of compounds "iridoid and iridoid glycosides"  in Catnip which attract the domestic animals specially the cats. I can recommend the following references

Lichman, B. R., Godden, G. T., Hamilton, J. P., Palmer, L., Kamileen, M. O., Zhao, D., ... & O’Connor, S. E. (2020). The evolutionary origins of the cat attractant nepetalactone in catnip. Science advances, 6(20), eaba0721.‏

Tagawa, M., & Murai, F. (1980). A new iridoid glucoside, nepetolglucosylester from Nepeta cataria. Planta medica, 39(06), 144-147.‏

Xie, S., Uesato, S., Inouye, H., Fujita, T., Murai, F., Tagawa, M., & Shingu, T. (1988). Absolute structure of nepetaside, a new iridoid glucoside from Nepeta cataria. Phytochemistry, 27(2), 469-472.‏

Thank you very much for these suggestions. In accordance with your suggestions, we included in the Introduction part these literature data.

Once again, thank you very much for your valuable comments related to improving the manuscript text. 

Reviewer 4 Report

Chemical composition of various Nepeta cataria plant organs’ methanol extracts associated with the in vivo hepato-protective and antigenotoxic features, and molecular modeling investigations by Milena et al.

Abstract:

Herein, the Wistar rats’ liver intoxication with 29 CCl4 resulted in the generation of hepatotoxic radicals, rCCl3 ● , and rCCl3OO● – It will be better to write it in detailed form for better understanding than just using the chemical formula especially in the abstract.

Line 32-43:

“Examined FME, LME, and SME, in the concentrations of 200, 32 100, and 50 mg/kg of body weight, respectively, administered orally to Wistar rats before intoxica- 33 tion by means of CCl4 injection, exerted the most notable hepatoprotective properties, seen through 34 the liver redox status (namely rSOD, rTBARS, rCAT, and rGSH) and toxicity markers (viz. rAST, 35 rALT, rALP, and rγ-GT), as well as the most promising antigenotoxic features quantified using the 36 comet assay, significantly reducing the levels of rDNA damage in the liver.”

“Distinct features may 37 be attributed to quercitrin (8406.31 mg/g), chlorogenic acid (1647.32 mg/g), and quinic acid (536.11 38 mg/g), found within the FME, rosmarinic acid (1056.14 mg/g), and chlorogenic acid (648.52 mg/g), 39 occurring within the LME, and chlorogenic acid (1408.43 mg/g), the most abundant in SME, i.e. sec- 40 ondary metabolites likewise individually administered in terms of identical experimental setup 41 against CCl4, whose pharmacology in vivo was elucidated in silico by means of the structure-based 42 studies within rCAT, as a redox marker, and rTopIIα, an enzyme catalyzing the rDNA double- 43 strand break.”

This is just two sentence, which can be easily divided into 4-5 sentences. Moreover in the current format I am not able to understand the intention of authors clearly.

In short Abstract should be re-written.

Major points:

1. Why only methanol extraction was performed, only one type of extraction might limit the identification of unknown metabolites.

2. One of the main was to identify the complete set of metabolites present in the extract then why not to use untargeted metabolite analysis approach to find any unknown variants of same metabolites present in low amount.

3. One more way is to fractionate the crude extracts to identify the low titre molecules present.

Identifying and quantifying already described molecules from same plant species does not provide any novelity to the current study.

Minor points:

1. Author should provide in more details the crude extract preparation and details.

2. MS acquisition details should be provided in details.

3. Scheme 2: the structures are not very clearly visible, so it is advised to redraw the structure for better illustration.

Author Response

Reviewer 4

Dear Sir/Madam,

Thank you very much for reviewing the enclosed manuscript. All your comments and remarks were carefully considered and below you can find the list of responses to each of the raised questions. All your suggestions significantly improved the quality of the manuscript.

Abstract:

Point 1. Herein, the Wistar rats’ liver intoxication with 29 CCl4 resulted in the generation of hepatotoxic radicals, rCCl3 ● , and rCCl3OO● – It will be better to write it in detailed form for better understanding than just using the chemical formula especially in the abstract.

Dear Sir/Madam,

Thank you very much for this valuable remark. The sentence is now rewritten, according to your recommendation, as follows:

Herein, the Wistar rats’ liver intoxication with CCl4 resulted in the generation of trichloromethyl and trichloromethylperoxy radicals, causing the lipid peroxidation within the hepatocytes membranes (viz. hepatotoxicity), as well as the subsequent formation of aberrant rDNA adducts, and consequent double-strand break (namely genotoxicity).

Point 2. Line 32-43:

“Examined FME, LME, and SME, in the concentrations of 200, 32 100, and 50 mg/kg of body weight, respectively, administered orally to Wistar rats before intoxica- 33 tion by means of CCl4 injection, exerted the most notable hepatoprotective properties, seen through 34 the liver redox status (namely rSOD, rTBARS, rCAT, and rGSH) and toxicity markers (viz. rAST, 35 rALT, rALP, and rγ-GT), as well as the most promising antigenotoxic features quantified using the 36 comet assay, significantly reducing the levels of rDNA damage in the liver.”

“Distinct features may 37 be attributed to quercitrin (8406.31 mg/g), chlorogenic acid (1647.32 mg/g), and quinic acid (536.11 38 mg/g), found within the FME, rosmarinic acid (1056.14 mg/g), and chlorogenic acid (648.52 mg/g), 39 occurring within the LME, and chlorogenic acid (1408.43 mg/g), the most abundant in SME, i.e. sec- 40 ondary metabolites likewise individually administered in terms of identical experimental setup 41 against CCl4, whose pharmacology in vivo was elucidated in silico by means of the structure-based 42 studies within rCAT, as a redox marker, and rTopIIα, an enzyme catalyzing the rDNA double- 43 strand break.”

This is just two sentence, which can be easily divided into 4-5 sentences. Moreover in the current format I am not able to understand the intention of authors clearly.

In short Abstract should be re-written.

Dear Sir/Madam,

Thank you very much for these valuable suggestions. The sentences in the Abstract are now clear by means of understanding and simplified in terms of avoiding the abbreviations. The Abstract is now rewritten, according to your recommendation, as follows:

This report summarizes the chemical composition analysis of Nepeta cataria L. flowers, leaves, and stems methanol extracts (FME, LME, SME, respectively) as well as their hepatoprotective and antigenotoxic features in vivo and in silico. Herein, the Wistar rats’ liver intoxication with CCl4 resulted in the generation of trichloromethyl and trichloromethylperoxy radicals, causing the lipid peroxidation within the hepatocytes membranes (viz. hepatotoxicity), as well as the subsequent formation of aberrant rDNA adducts, and consequent double-strand break (namely genotoxicity). Examined FME, LME, and SME, administered orally to Wistar rats before the injection of CCl4, exerted the most notable pharmacological properties in the concentrations of 200, 100, and 50 mg/kg of body weight, respectively. Thus, extracts’ hepatoprotective features were determined by monitoring the catalytic activities of enzymes and the concentrations of reactive oxidative species, modulating the liver redox status. On the other hand, the necrosis of hepatocytes was assessed by means of catalytic activities of liver toxicity markers. Extracts’ antigenotoxic features were quantified using the comet assay. Distinct pharmacological properties features may be attributed to quercitrin (8406.31μg/g), chlorogenic acid (1647.32 mg/g), and quinic acid (536.11 μg/g), found within the FME, rosmarinic acid (1056.14 μg/g), and chlorogenic acid (648.52 μg/g), occurring within the LME, and chlorogenic acid (1408.43 μg/g), the most abundant in SME. Hence, the plant’s secondary metabolites were individually administered similar to extracts, upon which their pharmacology in vivo was elucidated in silico by means of the structure-based studies within rat catalase, as a redox marker, and rat topoisomerase IIα, an enzyme catalysing the rat DNA double-strand break. Conclusively, the examined N. cataria extracts in specified concentrations could be used in clinical therapy for the prevention of toxin-induced liver diseases.

Point 3. Why only methanol extraction was performed, only one type of extraction might limit the identification of unknown metabolites.

Thank you for this valuable question. We strongly agree with your observation that only one type of extraction might limit the identification of secondary metabolites. However, due to the complexity of biological examinations of obtained extracts, as well as for reducing the number of experimental animals (more independent variables- solvents, type of extraction, etc… would result in a large number of sacrificed experimental animals), we used methanol only since it is a common and effective solvent for extraction of phenolics and flavonoids.

Point 4. One of the main was to identify the complete set of metabolites present in the extract then why not to use an untargeted metabolite analysis approach to find any unknown variants of the same metabolites present in low amount

Thank you very much for this question. Our current experimental facilities and the lack of adequate equipment for untargeted analysis (LC-HRMS) do not allow us to perform such experiments.

Point 5. One more way is to fractionate the crude extracts to identify the low titre molecules present.

Thank you very much for this valuable suggestion. It will be the further step in the examination of this plant and its secondary metabolites. In many cases, low abundant compounds exert significant biological activity. However, we cannot agree with the comment “Identifying and quantifying already described molecules from same plant species does not provide any novelty to the current study.”, because we corelated results of identification and quantification of phenolics and flavonoids with results of hepatoprotective and antigenotoxic activities.

Point 6. Author should provide in more details the crude extract preparation and details.

Thank you very much for this observation. For researchers from the area of Natural product chemistry this procedure is absolutely repeatable. Of course, a further in-depth description may lead to overlapping with sentences/words from other manuscript texts, leading to the violations of copyright issues and even plagiarism.

Point 7. MS acquisition details should be provided in details.

Thank you very much for this valuable suggestion. The MS acquisition data which we used were the same as MS acquisition parameters in references 61 and 62:

  1. Orcic, D.; Franciskovic, M.; Bekvalac, K.; Svircev, E.; Beara, I.; Lesjak, M.; Mimica-Dukic, N. Quantitative determination of plant phenolics in Urtica dioica extracts by high-performance liquid chromatography coupled with tandem mass spectrometric detection. Food Chem. 2014, 143, 48-53. Doi.org/10.1016/j.foodchem.2013.07.097
  2. Vukić, M.D.; Vuković, N.L.; Djelić, G.T.; Obradović, A.; Kacaniova, M.M.; Marković, S.; Popović, S.; Baskić, D. Phytochemical analysis, antioxidant, antibacterial and cytotoxic activity of different plant organs of Eryngium serbicum L. Ind. Crops Prod. 2018, 115, 88-97. doi.org/10.1016/j.indcrop.2018.02.031

So, in order to avoid any repetition of data which were published in previous manuscripts, we just cited these literature data.

 Point 7. Scheme 2: the structures are not very clearly visible, so it is advised to redraw the structure for better illustration.

Thank you very much for this valuable suggestion. The mechanism of action of rat catalase is rather complex and we tried to summarize it by the considered figure in terms of incorporating all the relevant chemical entities included in this multi-step reaction. Therefore, we find this an optimal way of presenting the process. However, to enhance the visibility, we embedded the figure with a better resolution of 600 dpi.

Round 2

Reviewer 2 Report

Dear Author, 

I would like to start with a small detail which, for me, makes a big difference. The fact that the number of bioethical  agreement from the local ethics comity "is unavailable" makes me rationally thinking that it is no approval. So in my point of view this is an unethical study, it would be illegal in many countries. In addition there are so many draw-backs in experimental protocol not only in the area of the 3Rs but also in research methodology which rise serios questions about relevance of the study. This is why I recommend the rejection of the manuscript. 

Author Response

Reviewer 2

Dear Author, 

I would like to start with a small detail which, for me, makes a big difference. The fact that the number of bioethical agreements from the local ethics comity "is unavailable" makes me rationally think that it is no approval. So in my point of view, this is an unethical study, it would be illegal in many countries. In addition, there are so many drawbacks in experimental protocol not only in the area of the 3Rs but also in research methodology which raise serious questions about the relevance of the study. This is why I recommend the rejection of the manuscript. 

Dear Sir/Madam,

Thank you very much for reviewing the enclosed manuscript. All your comments and remarks were carefully considered and below you can find the list of responses to each of the raised questions. All your suggestions significantly improved the quality of the manuscript.

Regarding the Ethical Committee, it gave the approval for conducting the experiments on experimental animals retroactively. To elaborate more on the matter, we hereby enclose the intercommunication with the Editor alongside a reasonable explanation for the lack of the Number of Ethical Approval for the initial experiments:

'' Dear Miss Lavinia Dumitrela Cozma,

Thank you very much for the email and for kindly asking the additional elaboration.

Hereby we provide the clarification for the date of the Ethical Approval.

Given that the experimental work regarding the animals for herein paper was conducted in 2021, the experimental work with animals was approved by the Ethical Committee of the Faculty of Science, the University of Kragujevac, which then acted and now acts following the relevant Serbian guidelines, including the Guidelines for the Care and Use of Laboratory Animals and Law on Animal Welfare ("Official Gazette of Republic of Serbia", No. 810 41/09) and the European Directive for the Welfare of Laboratory Animals Directive 2010/63/EU, for which the Number of Ethical Approval was not obligatory by Serbian laws. Therefore, the experiments were performed following the well-accredited guidelines by all means.

Following the same guidelines, some of the co-authors of the herein manuscript recently published several papers in high reputable journals, including the one of MDPI.

Kurtanović N, Tomašević N, Matić S, Mitrović MM, Kostić DA, Sabatino M, Antonini L, Ragno R, Mladenović M. Human estrogen receptor α antagonists, part 2: Synthesis driven by rational design, in vitro antiproliferative, and in vivo anticancer evaluation of innovative coumarin-related antiestrogens as breast cancer suppressants. Eur. J. Med. Chem. 2022, 227, 113869. doi: 10.1016/j.ejmech.2021.113869.

Kurtanović, N.; Tomašević, N.; Matić, S.; Proia, E.; Sabatino, M.; Antonini, L.; Mladenović, M.; Ragno, R. Human Estrogen Receptor Alpha Antagonists, Part 3: 3-D Pharmacophore and 3-D QSAR Guided Brefeldin A Hit-to-Lead Optimization toward New Breast Cancer Suppressants. Molecules 2022, 27, 2823. https://doi.org/10.3390/molecules27092823

Following your guidelines, we asked the Ethical Committee of the Faculty of Science, University of Kragujevac to review the paper and give the Number of Ethical Approval, based on the conducted studies. Given that the Number of Ethical Approval could not have been assigned retroactively, the Ethical Committee of the Faculty of Science, University of Kragujevac gave the approval that all the experiments were performed following the specified guidelines on June 30th 2022.

We hope that this clarification is now sufficient and that the paper will be again considered for publication in the Plants journal. We would like to stress again that all the experiments were performed following up-to-date Ethical Guidelines valid in the Republic of Serbia in 2021, externally validated by means of subsequent Number of Ethical Approval being provided upon request.

With kind regards,

Dr. Nenad Vuković ‘‘

The Editor replied on this letter as follows:

Decision

Accept after minor revision

Comments

The authors have provided reasonable justification for the ethical statement based on their institutional rule. Three reviewers evaluated this manuscript as acceptable. Upon my reading, I found this article suitable for Plants. However, the authors should carefully check the significant figures in the tables.

Regarding the experimental protocols, all experimental protocols were conducted correctly, both from scientific and ethical points.

Once again, thank you very much for your valuable comments related to improving the manuscript text.  

Reviewer 4 Report

The Manuscript "Chemical composition of various Nepeta cataria plant organs’ methanol extracts associated with the in vivo hepatoprotective and antigenotoxic features, and molecular modeling investigations" by Milena et al., has improved substantially after the careful revision by the authors, and hence, I recommend it for publication in the current form.